# Optimal, Active, Prediction-powered AI Model Evaluation on a Budget

## Abstract

The development lifecycle of generative AI systems requires continual evaluation, data acquisition, and annotation, which is costly in both resources and time. In practice, a desire for rapid iteration often makes it necessary to rely on synthetic annotation data because of its low cost, despite the potential for substantial bias. In this paper, we develop a rigorous theoretical framework for novel, cost-aware evaluation pipelines that actively balance the use of a cheap, but often inaccurate, weak rater—such as a model-based *autorater* that is designed to automatically assess the quality of generated content—with a more expensive, but also more accurate, strong rater such as a human annotator. Building on recent work in active and prediction-powered statistical inference, we theoretically derive a family of cost-optimal policies for allocating a given annotation budget between weak and strong raters so as to maximize statistical efficiency. Next, using synthetic and real-world data, we empirically characterize conditions under which these types of policies can yield significant improvements over classical methods. Finally, we find that practical approximations of the theoretically optimal policies can achieve the same estimation precision at a far lower total annotation budget than standard evaluation methods, especially in tasks where there is high variability in the difficulty of examples.

## 1 Introduction

Accurately and efficiently evaluating generative AI (GenAI) systems is a core technical challenge, both for model development and for reliable model deployment. In this paper, we introduce new statistical tools for active, cost-sensitive model evaluation. Specifically, we develop evaluation pipelines that dynamically annotate data using a mix of weak and strong annotation options in a way that is aware of their relative costs and strengths. The core idea is to strategically balance inexpensive but potentially inaccurate annotations from a *weak rater* against more accurate, but also more costly, annotations from a more sophisticated *strong rater* alternative. Our goal will be to use the weak raters to help give unbiased estimates of the mean of the strong rater's judgments. This is a key target for many AI evaluation applications, as it captures fundamental metrics like model accuracy, win-rate, or hallucination rate. In this work, we study this problem both from a theoretical perspective, in which we gain a rigorous understanding of what it takes to do so in a cost-optimal manner, and also from an applied perspective, in which we develop effective approximations of the cost-optimal strategies to use in practice.

The exact composition of the weak and strong raters is flexible; for example, the weak rater might be a small AI model or rule-based heuristic, while the strong rater might be a larger AI model, an AI model with tools or larger inference-time reasoning capabilities, a human, or even the consensus of multiple expert humans. The cost of the evaluation might then be measured in compute, latency, or dollars. Active evaluation aims to minimize cost by selectively obtaining expensive annotations only when they are informative, relying on the cheaper option otherwise. All of the annotations are then combined using statistically principled, unbiased methods to yield reliable, yet cost-effective, performance metrics.

Combining different data sources to improve evaluation quality is not new: in particular, the use of cheap but biased metrics as control variates to improve statistical efficiency in model evaluation has been explored before from various perspectives (Angelopoulos et al., 2023a;b; Boyeau et al., 2024; Chaganty et al., 2018; Chatzi et al., 2024; Fisch et al., 2024; Jung et al., 2025; Saad-Falcon et al., 2024; Zrnic & Candès, 2024). Here, our main technical contribution is a theoretical framework for active, prediction-powered model evaluations that strategically choose when to deploy the strong

rater as opposed to the weak rater in order to maximize evaluation accuracy subject to a budget on the expected total cost. Informally, these policies solve the following constrained optimization problem:

$$\text{maximize} \quad \text{Accuracy of the evaluation,}$$
$$\text{subject to} \quad \text{Cost of the evaluation remaining below a budget } B.$$

We derive these optimal policies via new technical extensions and combinations of modern techniques in statistics, namely, active statistical inference (Zrnic & Candès, 2024) and prediction-powered inference (PPI; Angelopoulos et al., 2023a;b; Zrnic & Candès, 2024). As we will prove, the resulting oracle policies—**which represent the best strategies we can hope to achieve in theory**—depend on (i) the rater costs, but also on (ii) task-specific distributional properties (like the weak rater's error) that are *often unknown in practice*. That said, **using the form of these optimal policies as a guiding foundation, we test and analyze empirical approximations** that operate by first estimating these unknown quantities from data (e.g., using a "burn-in" set), and then use the theoretical form of the optimal policy with the estimated parameters plugged in (and provide bounds on the optimality gap). **Empirically, we demonstrate that this practical approach can achieve substantial savings over passive strategies**, although we also highlight important open challenges for future work that naturally arise due to "cold-start" issues, as well as imperfections of existing autorater models and their uncertainty estimates.

Finally, though AI model evaluation is the primary motivation and focus in this paper, we note that our framework also extends to general convex M-estimation problems (in any domain). See Appendix B.

**Related work.** Prediction-powered inference (PPI; Angelopoulos et al., 2023a;b; Zrnic & Candès, 2024) is the technique of combining a small number of trusted observations with predictions from a machine learning system for the purpose of statistical estimation. Its core statistical principles are closely related to control variate estimators (Chaganty et al., 2018; Ripley, 1987) as well as semi-parametric inference with missing data (Chernozhukov et al., 2018; Robins & Rotnitzky, 1995; Tsiatis, 2006). Recently, a body of work has explored applying PPI to the evaluation of GenAI systems, where human annotations are combined with "autorater" outputs (Boyeau et al., 2024; Chatzi et al., 2024; Egami et al., 2023; Fisch et al., 2024; Saad-Falcon et al., 2024); though it has also been noted that the sample efficiency gained is limited when the autorater is not sufficiently accurate (Dorner et al., 2025; Thakur et al., 2025). A natural extension of PPI is to actively select a fixed number of examples on which to obtain trusted observations, while deferring the remaining examples to the autorater (Gligorić et al., 2024; Zrnic & Candès, 2024). Roughly speaking, these approaches sample human annotations with probability proportional to the uncertainty of the autorater. However, they work only in a restricted setting in which the ratio of expensive to cheap ratings, $n/N$, is fixed in advance, and then pick the optimal policy subject to that constraint. No guidance is given as to what this ratio should be based on the relative costs of the ratings, or even what the total number of examples $N$ should be.

**Contributions.** Our work extends this literature both theoretically and empirically. Our core theoretical contribution is the derivation of error-minimizing sampling rules under cost constraints. That is, previous methods decide in advance some fixed ratio of cheap to expensive ratings $n/N$ and a policy that maximizes accuracy under that fixed ratio, while our policy maximizes accuracy subject to a general total cost constraint; as such the ratio of cheap to expensive ratings in our methods is not fixed in advance, but determined by the solution to the overall optimization problem. We theoretically derive two forms of optimal policies: (i) the best fixed sampling rate (Proposition 1), and (ii) the best active sampling rule that depends on covariates (Proposition 2). We will see that both of these policies depend on the *cost ratio* between cheap and expensive ratings, and certain measures of how "worth the cost" it is to query the expensive rater versus the cheap rater. One additional novelty of our work is that it improves upon the policy proposed by Zrnic & Candès (2024) by accounting for the constraint that the policy must lie in $[0, 1]$ for all values of $x$. Finally, Appendix B includes further theoretical innovations, such as an extension to convex M-estimators and an optimal method for selecting the covariate $x$ (as opposed to only the label, as considered in the prior work).

On the empirical end, we extend the scope of the standard PPI framework to heterogeneous model evaluation settings involving two distinct rating sources, each with a different cost-performance profile. This goes beyond the typical "human-vs-LLM" scenario described above, and encompasses any situation where less expensive, less accurate ratings are combined with more expensive, more accurate ones, even if both sources are automated (e.g., smaller vs. larger modelsm, or more vs. less inference-time reasoning). In Sections 3 and 4, we present an extensive empirical investigation into the conditions under which these new sampling rules prove beneficial over classical estimation. Specifically, we

identify that the success of our framework is determined by: (a) the overall error of the weak rater, (b) the overall variance of the target strong rater, and (c) the heteroskedasticity of the weak rater's errors.

## 2 COST-OPTIMAL ANNOTATION POLICIES

We now describe our methods for constructing active, cost-optimal model evaluations. The methods rely on one critical ingredient: an *annotation policy* $\pi$. The job of the annotation policy is to look at the input and decide whether it should be labeled by the expensive rater. The theory in this section derives the **optimal policies under different restrictions on the policy space**. These policies are *oracle* policies—we prove that they depend on properties of the data distribution, some of which are impossible to know in advance. As described in Section 1, the point of this section is to tell us **what kinds of policies we should be targeting, not how to find them**; later, we will explore how to estimate them in practice. Proofs of all theoretical results can be found in Appendix C.

### 2.1 BASIC NOTATION

We observe inputs $X \sim P_X$ from some space $\mathcal{X}$ and distribution $P_X$: in the setting of LLMs, we think of the input $X$ as containing the prompt as well as the response from one or multiple LLMs. Our goal is to approximate an expensive rating $h(X) \in \mathbb{R}$, such as a human preference, with a cheap automated evaluator $g(X) \in \mathbb{R}$; for notational convenience we define $H \triangleq h(X)$ and $G \triangleq g(X)$. In our setup, querying $H$ and $G$ cost $c_h$ and $c_g$, respectively. We seek to query $H$ only when it is "worth the cost".

We consider a sequential setting: for every $t \in \mathbb{N}$, we observe i.i.d. $X_t \sim P_X$ and $G_t \sim P_{G|X}$. Upon observing $X_t$, we then have the option to query $H_t \sim P_{H|X}$. Our objective is to estimate $\theta^* = \mathbb{E}[H]$, the mean target rating. To this end, we develop estimators that efficiently sample only the data points for which $H_t$ is needed, and stop sampling after a certain budget is exhausted. Define the random variable $\xi_t \sim \text{Bern}(\pi_t(X_t))$, which is the indicator of whether we sampled $H_t$. It equals 1 with probability $\pi_t(X_t)$, and we have the freedom to choose the annotation policy $\pi_t$ based on the previous data we have seen so far. We estimate $\theta^*$ with the following estimator, defined for all $T \in \mathbb{N}$:

$$\hat{\theta}_T = \frac{1}{T} \sum_{t=1}^{T} \Delta_t \quad \text{where } \Delta_t = G_t + (H_t - G_t) \frac{\xi_t}{\pi_t(X_t)}. \tag{1}$$

It is easy to show that $\hat{\theta}_T$ is unbiased, with $\mathbb{E}[\hat{\theta}_T] = \theta^*$; see Appendix B.1. Here $\pi_t \in \Pi$ for some policy class $\Pi$. If $\Pi$ is left unspecified, it should be assumed that $\pi_t$ can be any function with range $(0, 1]$. This is the sequential estimator from Zrnic & Candès (2024): the difference will be in how we set $\pi_t$ to balance labeling costs. In general, the annotation policy $\pi_t$ is allowed to change arbitrarily online as a function of past data, as is the predictor $g$. For simplicity, we will focus on the setting where the parameters of $\pi$ and $g$ remain fixed throughout and are *not* updated online, or as if we are updating in batches; however our results will also hold asymptotically when $\pi$ and $g$ are updated online and converge. We use the notation $\hat{\theta}_T^\pi$ to denote the estimator in (1) with a fixed policy $\pi$, i.e., where $\pi_t = \pi, \forall t \in T$.

To calculate the cost and error of our estimator, we additionally define:

$$\text{Error}_T(\pi) \triangleq \mathbb{E}\left[\left(\hat{\theta}_T^\pi - \theta^*\right)^2\right] = \frac{1}{T}\left(\text{Var}(H) - \mathbb{E}[(H - G)^2] + \mathbb{E}\left[(H - G)^2 \frac{1}{\pi(X)}\right]\right), \tag{2}$$

and

$$\text{Cost}_T(\pi) \triangleq T(c_h \mathbb{E}[\pi(X)] + c_g).$$

See Appendix B.2 for a short derivation. These functions describe the mean squared error and expected cost of the estimator with annotation policy $\pi$ as a function of time, and our goal will be to minimize one subject to a constraint on the other. When we refer to a budget on the cost, it will be denoted as $B$. Furthermore, we note that, for convenience, the cost-optimized policies that we present in the remainder of this section will relax the constraint that the stopping time $T^{\text{stop}}$ at which $\text{Cost}_{T^{\text{stop}}}(\pi)$ is just under budget must be an integer, though this does not have a significant effect on the optimization for large enough budgets $B$ where $T^{\text{stop}} \gg 1$.

## 2.2 Optimal random annotation

The simplest annotation policy does not depend on $X$, and simply queries $H$ with some fixed probability, which we denote as $\pi(x) = p$ for a sampling rate $p \in (0, 1]$. In other words, we let $\pi \in \Pi^{\mathrm{random}} = \{x \mapsto p : p \in (0, 1]\}$. When $p$ is too large, the cost is too high; when $p$ is too small, the error blows up. Our job is to choose the optimal balance, and the next result shows it has a simple, explicit form that depends on the cost ratio $c_g/c_h$ and the error of $G$ compared to the variance of $H$.

**Proposition 1.** *Let* $(X_1, G_1, H_1), \ldots, (X_T, G_T, H_T)$, $T \in \mathbb{N}$, *be an i.i.d. sequence of real-valued random variables with joint distribution $P$, and define* Error, Cost, *and* $\Pi^{\mathrm{random}}$ *as above. Assume that* $\mathbb{P}(G_1 = H_1) < 1$ *and that* $c_h > c_g > 0$, *and define the optimization problem*

$$\underset{\pi \in \Pi^{\mathrm{random}},\, T^{\mathrm{stop}} \in \mathbb{R}_{>0}}{\text{minimize}} \quad \mathsf{Error}_{T^{\mathrm{stop}}}(\pi) \quad \text{subject to} \quad \mathsf{Cost}_{T^{\mathrm{stop}}}(\pi) \leq B. \tag{3}$$

*Then the solution to Problem (3) for all $x \in \mathcal{X}$ is*

$$\pi_{\mathrm{random}}(x) = \begin{cases} \sqrt{\dfrac{c_g}{c_h} \dfrac{\mathbb{E}[(H-G)^2]}{\mathrm{Var}(H) - \mathbb{E}[(H-G)^2]}} & \text{if } \mathbb{E}[(H-G)^2] < \dfrac{c_h}{c_h + c_g}\mathrm{Var}(H) \\ 1 & \text{otherwise.} \end{cases} \tag{4}$$

We can make a few observations about $\pi_{\mathrm{random}}$. First, if the mean squared error of the weak rater $G$ is greater than the variance of $H$ (or more precisely, more than a $c_h/(c_h + c_g)$ fraction of the variance of $H$), then it is not helpful—and we should simply choose to query $H$ all the time. If $\mathrm{MSE}(H, G)$ is sufficiently low, however, then the rate at which we sample $H$ varies inversely with both the ratio of $\mathrm{Var}(H)$ to $\mathrm{MSE}(H, G)$ and the ratio of the cost of $H$ to the cost of $G$. This makes intuitive sense: if the target label $H$ is high variance but our "weak" rater $G$ is in fact a fairly "strong" rater (in that it produces similar ratings to those of $H$), then we should primarily exploit $G$'s low cost, high-quality predictions, while sampling $H$ at just a low rate to correct for any minor bias that arises.

## 2.3 Optimal active annotation

Next, we study policies that *depend* on $X$; i.e., they query $H$ with some probability that depends on $X$. This strategy can greatly improve statistical power when the error distribution is heteroskedastic in $X$; for example, when some prompts are much harder than others. In this setting, it makes sense for $\pi$ to depend on $X$, and to ask for advanced rating help more often when $G$ is likely to be wrong. Towards that end, we define our annotation policy class to be $\pi \in \Pi = \{x \mapsto f(x) : f(x) \in (0, 1]; \forall x \in \mathcal{X}\}$, which is the set of annotation policies placing a strictly positive amount of sampling mass on each query. As the next proposition shows, the optimal policy in this setting will depend on the uncertainty of the weak rater, $u(x) \triangleq \mathbb{E}[(H-G)^2 \mid X = x]$, expressed as the expected mean squared conditional error given $X = x$. For notational convenience, we also define the random variable $U \triangleq u(X)$.

**Proposition 2.** *In the same setting as Proposition 1, define $\Pi$ as above, let $\mathcal{X}$ be discrete, and additionally define the optimization problem*

$$\underset{\pi \in \Pi,\, T^{\mathrm{stop}} \in \mathbb{R}_{>0}}{\text{minimize}} \quad \mathsf{Error}_{T^{\mathrm{stop}}}(\pi) \quad \text{subject to} \quad \mathsf{Cost}_{T^{\mathrm{stop}}}(\pi) \leq B. \tag{5}$$

*Define the scaled and clipped policy, $\pi_{\mathrm{clip}}$, as:*

$$\pi_{\mathrm{clip}}(x; \tau) = \min\left(\gamma^*(\tau)\sqrt{u(x)}, 1\right) = \begin{cases} \gamma^*(\tau)\sqrt{u(x)} & \text{if } \sqrt{u(x)} \leq \tau \\ 1 & \text{otherwise,} \end{cases}$$

*where $c_h > c_g > 0$ and $\gamma^*(\tau) \in \left(0, \frac{1}{\tau}\right]$ is defined as*

$$\gamma^*(\tau) = \min\left(\sqrt{\dfrac{c_g/c_h + \mathbb{P}\left(U > \tau^2\right)}{\left(\mathrm{Var}(H) - \mathbb{E}[U\mathbb{1}\left\{U \leq \tau^2\right\}]\right)_+}}, \dfrac{1}{\tau}\right).$$

*Then the solution to Problem (5) is $\pi_{\mathrm{active}}(x) = \pi_{\mathrm{clip}}(x; \tau^*)$, where $\tau^* > 0$ is the solution to*

$$\tau^* = \underset{\tau \in \mathbb{R}_{>0}}{\arg\min} \left(c_h \mathbb{E}[\pi_{\mathrm{clip}}(x; \tau)] + c_g\right)\left(\mathrm{Var}(H) + \mathbb{E}\left[U\left(\pi_{\mathrm{clip}}(x; \tau)^{-1} - 1\right)\right]\right).$$

**Remark 3.** *The final optimization problem presented for the clipping threshold $\tau^*$ is non-convex and has no analytical solution. However, because it is a 1-dimensional optimization problem, we can coarsely discretize and optimize $\tau$ via simple grid search. See also Appendix D.5.*

On a technical level, the solution in Proposition 2 has a similar form to the active sequential estimator proposed in Zrnic & Candès (2024), but with an optimized proportionality constant, as well as additional clipping to rigorously account for the constraints on $\pi(x) \in (0, 1]$. The latter point is particularly important, as it is not accounted for in prior work. In contrast to the fixed, prespecified ratio prescribed by prior work, in Appendix B.8 we show how the *cost-optimal* target ratio of expensive to cheap ratings can be as extreme as 0 or 1, depending on the cost ratio of $G$ to $H$.

While the form of $\pi_{\text{active}}$ is more complex than that of $\pi_{\text{random}}$, it still admits a fairly straightforward interpretation: for some confidence threshold $\tau^*$ below which the conditional mean squared error of $G$ over all confident data points with $\sqrt{u(x)} \leq \tau^*$ is sufficiently low, we sample proportional to $\sqrt{u(x)}$. On the remaining highly uncertain examples where $\sqrt{u(x)} > \tau^*$, we always use $H$, and ignore $G$. The exact threshold $\tau^*$ depends on the distributions of $H$ and $G$, and their cost-ratio.

We can also observe that Proposition 2 is a direct generalization of Proposition 1. When $X$ is independent of $(H - G)^2$ so that $u(x) = \mathbb{E}[(H - G)^2] \, \forall x \in \mathcal{X}$, the policy $\pi_{\text{active}}$ reduces to $\pi_{\text{random}}$:

$$\underbrace{\gamma^*(\tau^*)\sqrt{u(x)}}_{\text{optimal active}} = \sqrt{\frac{c_g}{c_h} \frac{\mathbb{E}[(H - G)^2 \mid X = x]}{\text{Var}(H) - \mathbb{E}[(H - G)^2]}} = \underbrace{\sqrt{\frac{c_g}{c_h} \frac{\mathbb{E}[(H - G)^2]}{\text{Var}(H) - \mathbb{E}[(H - G)^2]}}}_{\text{optimal random}}.$$

The intuitive conclusion is that active querying can help **if the conditional squared error of $G$ has significant variance** to it (i.e., there exist some regions of $\mathcal{X}$ where $G$ has a much higher level of agreement with $H$ than on other regions of $\mathcal{X}$, such as on easy vs. hard examples). This can be contrasted with the optimal random policy, $\pi_{\text{random}}$, from (4): there we sample at a fixed rate for each $X$, where that rate depends only on $G$'s *average* error with respect to $H$ across all types of inputs.

> **Takeaways: Cost-optimal annotation policies**
>
> We derive two policies for sampling the expensive rating $H$ given a budget $B$: $\pi_{\text{random}}$ chooses the optimal fixed probability $p^* \in (0, 1]$, while $\pi_{\text{active}}$ defines an optimal input conditional probability $\pi_{\text{active}}(x) \in (0, 1]$. Both navigate the following trade-off: reducing $\mathbb{E}[\pi(X)]$ **increases the total number of samples we can afford to rate at all**, but not querying $H$ when $G$ is inaccurate **increases variance**. Finally, both policies converge to the baseline estimator (i.e., $\pi_{\text{base}}(x) = 1$) when the error of $G$ is too high relative to the variance of $H$.

## 3 COMPARING COST-OPTIMAL POLICIES IN SIMULATED SETTINGS

The estimation error of the optimal policies presented in Section 2 depends on the distributions of the expensive target label $H$, the cheap estimated label $G$, and the cost-ratio $c_g/c_h$ for querying $G$ versus $H$. To build a clearer understanding of how these variables influence the performance of our proposed policies, we now conduct a series of carefully controlled experiments on simulated data. Note that since all of the key distributional quantities (i.e., $\text{Var}(H)$, $\text{MSE}(H, G)$, etc) are *known* in the synthetic settings we consider in this section, we are also able to compute $\pi_{\text{active}}$ and $\pi_{\text{random}}$ exactly—as opposed to the more difficult real-world data settings we will tackle next in Section 4.

### 3.1 METRICS

To measure the relative performance of annotation policy $\pi_1$ vs $\pi_2$, we compute the *ratio* of their errors at $T_i^{\text{stop}}$. Once again relaxing the restriction that $T_i^{\text{stop}} \in \mathbb{N}$, we compute a budget-free

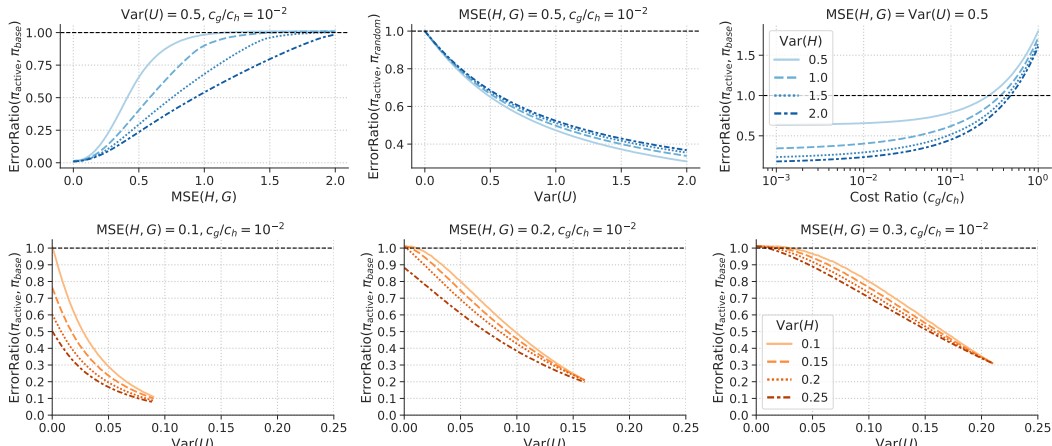

Figure 1: Results on the Gaussian (top) and Bernoulli (bottom) settings while varying $\text{MSE}(H, G)$, $\text{Var}(U)$, and $c_g/c_h$. Each line plots a different value of $\text{Var}(H)$, where we choose values that are representative of low, medium, or high variance settings compared to $\text{MSE}(H, G)$. In the Bernoulli setting, $\text{MSE}(H, G) = \mathbb{E}[H \neq G]$, and $\text{Var}(U)$ can be at most $\text{MSE}(H, G)(1 - \text{MSE}(H, G))$.

approximation based on the expression for $\text{Error}_{T_i^{\text{stop}}}(\pi)$, where $T_i^{\text{stop}} = B/(c_h \mathbb{E}[\pi_i(X)] + c_g)$:

$$\text{ErrorRatio}(\pi_1, \pi_2) \triangleq \frac{\left(c_h \mathbb{E}[\pi_1(X)] + c_g\right)\left(\text{Var}(H) - \mathbb{E}[(H - G)^2] + \mathbb{E}\left[(H - G)^2 \frac{1}{\pi_1(X)}\right]\right)}{\left(c_h \mathbb{E}[\pi_2(X)] + c_g\right)\left(\text{Var}(H) - \mathbb{E}[(H - G)^2] + \mathbb{E}\left[(H - G)^2 \frac{1}{\pi_2(X)}\right]\right)}.$$

Note that while $\text{ErrorRatio}(\pi_1, \pi_2)$ does not depend on the budget, it does implicitly depend on $P_X$ as well as $P_{H|X}$ and $P_{G|X}$. We will focus on $\text{ErrorRatio}(\pi_{\text{active}}, \pi_{\text{base}})$, the error ratio of the active estimator to the baseline estimator which only uses $H$, as well as $\text{ErrorRatio}(\pi_{\text{active}}, \pi_{\text{random}})$, which compares the active estimator to the estimator that doe not depend on $X$. Note that for $\text{ErrorRatio}(\cdot, \pi_{\text{base}})$, we disregard $c_g$ for $\pi_{\text{base}}$, and replace the denominator with $c_h \text{Var}(H)$.

## 3.2 GAUSSIAN DATA

We construct an experiment where we change $\text{Var}(H)$, $\text{MSE}(H, G)$, and $\text{Var}(U)$ independently (recall that we introduced $U \triangleq u(X) = \mathbb{E}[(H - G)^2 \mid X]$ in Section 2.3). First, we draw $H \sim \mathcal{N}(0, \nu)$ so that $\mathbb{E}[H] = 0$ and $\text{Var}(H) = \nu$. Then we draw $U \sim \text{Gamma}\left(\mu^2/\eta, \eta/\mu\right)$ so that $\text{MSE}(H, G) = \mathbb{E}[U] = \mu$ and $\text{Var}(U) = \eta$. Finally, we set $G = H + \sqrt{U}$.

Results are shown in the top row of Figure 1. The left panel plots the error ratio of $\pi_{\text{active}}$ to $\pi_{\text{base}}$ as a function of $\text{MSE}(H, G)$ and for different $\text{Var}(H)$, while keeping $\text{Var}(U) = 0.5$. As expected, the error of $\pi_{\text{active}}$ increases with the $\text{MSE}(H, G)$, with the rate of increase influenced by $\text{Var}(H)$. When $\text{MSE}(H, G)$ is large relative to $\text{Var}(H)$, $\pi_{\text{active}}$ provides no benefit over $\pi_{\text{base}}$. The middle panel plots the error ratio of $\pi_{\text{active}}$ to $\pi_{\text{random}}$ while varying $\text{Var}(U)$ for a fixed $\text{MSE}(H, G)$. For small values of $\text{Var}(U)$, the conditional error in $G$ is nearly the same everywhere, and there is no benefit to using $\pi_{\text{active}}$ over $\pi_{\text{random}}$. Larger values of $\text{Var}(U)$, however, lead to a performance advantage for $\pi_{\text{active}}$. The right panel plots the error ratio of $\pi_{\text{active}}$ to $\pi_{\text{base}}$ while keeping $\text{MSE}(H, G)$ and $\text{Var}(U)$ fixed, but varying $c_g/c_h$. As expected, $\pi_{\text{active}}$ is most effective when $c_g \ll c_h$.

## 3.3 BERNOULLI DATA

While the Gaussian setting above is informative, in many typical situations $H$ is bounded, such as when $H$ is a binary, Bernoulli rating for win-rate or accuracy estimation. This creates a more difficult setting for $\pi_{\text{active}}$, since both $\text{Var}(H)$ and $\text{Var}(U)$ are upper-bounded by $0.25$ for Bernoulli $H$. In fact, in binary settings, $\text{MSE}(H, G)$ and $\text{Var}(U)$ are in tension: the more accurate $G$ is, the lower the variance of its errors, and $\pi_{\text{active}}$ will be limited in terms of any relative benefit it can provide

over $\pi_{\text{random}}$. The same is also true for when $G$ is uniformly *inaccurate*. To better analyze this kind of setting, we construct a binary dataset where first we draw $H \sim \text{Bern}(0.5 + \sqrt{0.25 - \nu})$, so that $\text{Var}(H) = \nu$. Next, we draw $U$ from a Beta distribution with mean $\mu$ and variance $\eta$, where $\eta \leq \mu(1-\mu)$, which is satified by $U \sim \text{Beta}(\kappa\mu, \kappa(1-\mu))$ for $\kappa = \frac{\mu(1-\mu)}{\eta} - 1$. Finally, we flip $H$ with probability $U$ to get the prediction $G$ (i.e., $G$ is also Bernoulli with $\text{MSE}(H, G) = \mu$).

Results are shown in the bottom row of Figure 1 for $\pi_{\text{active}}$ vs $\pi_{\text{base}}$ (see Appendix D for $\pi_{\text{active}}$ vs. $\pi_{\text{random}}$). As in the Gaussian setting, the error ratio of $\pi_{\text{active}}$ to $\pi_{\text{base}}$ improves dramatically with larger $\text{Var}(U)$. Note that the active and random estimator are the same when $\text{Var}(U) = 0$. For larger $\text{MSE}(H, G)$, $\text{Var}(U)$ must also be increasingly large for $\pi_{\text{active}}$ to improve significantly over $\pi_{\text{base}}$. Indeed, on the right-hand side of the bottom row of Figure 1 where $\text{MSE}(H, G) > \text{Var}(H)$, we can see that $\pi_{\text{random}}$ provides no benefits over $\pi_{\text{base}}$; that is, $\text{ErrorRatio}(\pi_{\text{random}}, \pi_{\text{base}}) = 1$ when $\text{Var}(U) = 0$, which corresponds to the fixed-rate sampling policy as noted earlier. When $\text{Var}(U) \gg 0$, however, $\pi_{\text{active}}$ can obtain substantially lower estimation error than $\pi_{\text{base}}$. Still, unlike the earlier Gaussian data, the best active error ratio in this setting is bounded from below by $\text{MSE}(H, G)$, and is achieved when $U$ has maximum variance (which is also bounded).

> **Takeaways: Performance characteristics of cost-optimal annotation policies**
>
> In general, the following properties hold for active annotation versus standard annotation (similar findings for random): (i) as the error of $G$, $\text{MSE}(H, G)$, increases, the benefit **decreases**; (ii) as the variance of the conditional squared-error of $G$, $\text{Var}(U)$, increases, the benefit **increases**; and (iii) as the cost ratio, $c_g/c_h$, of $G$ relative to $H$ increases, the benefit **decreases**.

## 4  ESTIMATING COST-OPTIMAL POLICIES IN PRACTICE

The theoretical results in Section 2 derive optimal annotation policies under the assumption that the key distributional parameters governing the relationship between the expensive rater ($H$) and the cheap rater ($G$) are known. In reality, these parameters must be estimated (imperfectly; see Appendix B.6 for theoretical error analysis). Furthermore, the optimal threshold $\tau^*$ and scaling factor $\gamma(\tau^*)$ for the active policy $\pi_{\text{active}}$ in Proposition 2 also depend on conditional versions of these unknown quantities (e.g., the conditional MSE, $\mathbb{E}[(H - G)^2 \mid U \leq \tau]$). Some of these estimates can be derived automatically from the model itself, for example if $g(x) \in [0, 1]$ is a binary classifier, we may choose $u(x) = g(x)(1 - g(x))$, which is equal to $\mathbb{E}[(H - g(x))^2 \mid X = x]$ when $g(x) = \mathbb{P}(H = 1 \mid X = x)$. For simplicity, this is the approach we take in our experiments. That said, if $g(x)$ is not calibrated, $u(x)$ will not be accurate. In that case, many other ways of obtaining $u(x)$ exist—we leave the exploration of these alternative methods, such as by asking an LLM directly for its confidence (Kadavath et al., 2022; Xiong et al., 2024), to future work. For the key parameters $\text{Var}(H)$, $\text{MSE}(H, G)$, $\gamma(\tau^*)$, and $\tau^*$, we explore estimating them using the following approaches:

**Policy transfer from related datasets (A1).** Here we *transfer* all parameters necessary for $\pi$ from a separate, but related, dataset. For example, in Section 4.2, we use data from the Chatbot Arena dataset (Zheng et al., 2023; Chiang et al., 2024) to estimate the win-rate of GPT-4 over Claude 2.1, but transfer parameters for $\pi_{\text{random}}$ and $\pi_{\text{active}}$ from a separate set of comparisons between different available models. We also calibrate $G$ using Platt scaling (Platt, 1999) on the transfer dataset.

**Policy burn-in on the first $n_b$ examples (A2).** When a suitable transfer dataset is not available as in A1, we can take a hybrid approach where we start by sampling $H$ for the first $n_b = 200$ examples[1] with probability 1, and then use them to estimate the parameters necessary for $\pi_{\text{active}}$ and $\pi_{\text{random}}$. We also calibrate $G$ using Platt scaling on these $n_b$ examples. As a fair comparison to the baseline method of only using $H$, we allow these $n_b$ samples to also be used as data for estimating $\theta = \mathbb{E}[H]$. Specifically, we use the (estimated) inverse-variance-weighted average of the annotation policy $\pi$'s estimate, $\hat{\theta}_T^\pi$, and the classical estimate on the burn-in data, $\hat{\theta}_{n_b} = \frac{1}{n_b} \sum_{i=1}^{n_b} H_i$,

$$\hat{\theta}_{T^{\text{stop}}+n_b}^\pi = \frac{\widehat{\text{Var}}(\hat{\theta}_{T^{\text{stop}}}^\pi)}{\widehat{\text{Var}}(\hat{\theta}_{n_b}) + \widehat{\text{Var}}(\hat{\theta}_{T^{\text{stop}}}^\pi)} \hat{\theta}_{n_b} + \frac{\widehat{\text{Var}}(\hat{\theta}_{n_b})}{\widehat{\text{Var}}(\hat{\theta}_{n_b}) + \widehat{\text{Var}}(\hat{\theta}_{T^{\text{stop}}}^\pi)} \hat{\theta}_{T^{\text{stop}}}^\pi. \tag{6}$$

We estimate $\widehat{\text{Var}}(\cdot)$ on the burn-in data. Note that $\hat{\theta}_{T^{\text{stop}}+n_b}$ is asymptotically unbiased in this case.

---

[1] See Appendix D.4 for ablations on the effects of larger or smaller $n_b$; small $n_b$ is typically sufficient.

Unlike our earlier experiments in Section 3, because we are now using policies with *estimated* parameters, they are no longer guaranteed to be optimal—though as we will see in Section 4.3, they can still be quite effective empirically when using either estimation approach (A1 or A2). To get a sense of how close to optimal the estimated policies are, we also compute an **Oracle**: $\pi_{\text{active}}$ with parameters computed using the whole dataset, and $u(x)$ taken directly as $|h(x) - g(x)|^2$.

## 4.1 METRICS

We compare the baseline method $\pi_{\text{base}}$ of always sampling $H$ with the random policy $\pi_{\text{random}}$ and the active policy $\pi_{\text{active}}$. For each policy, we compute the **mean squared error**, $\mathbb{E}[(\hat{\theta}_T^\pi - \theta^*)^2]$, for a range of budgets $B$ ($c_h$ is normalized to be one "cost unit"), with 95% bootstrap CIs shown over $15k$ trials. We then compute the **mean effective budget**, which we define as the budget $B'$ required for $\pi_{\text{base}}$ to achieve the same MSE as the given policy $\pi$ at a budget $B$. If $\pi$ is more cost-effective than $\pi_{\text{base}}$, then $B'$ will be larger than $B$ (higher is better). Finally, we also compute the **mean cost savings** for a given mean-squared error, which we define as the budget deficit relative to $\pi_{\text{base}}$ required to achieve that target error (higher is better). By definition, we have that the mean effective budget for $\pi_{\text{base}}$ is the line $y = x$ (since $B' = B$ always), while the cost savings for $\pi_{\text{base}}$ is 0.

## 4.2 DATASETS

We report results on three datasets, which span a diverse range of raters and distributional characteristics. For each task, we calculate $\theta^* = \mathbb{E}[H]$ using the full dataset. For simplicity we assume that the total number of data points $X_t$ is at least $\lceil B/c_g \rceil$, and sample with replacement from the original data if not. We leave treatment of datasets where $T^{\text{stop}} \leq T^{\text{max}}$ (and the constraint is active) to future work. See Appendix D for results on two additional datasets, ImageNet (Deng et al., 2009) and Seahorse (Clark et al., 2023), as well as other ablations and analysis.

**Chatbot Arena.** The Chatbot Arena dataset (Zheng et al., 2023; Chiang et al., 2024) evaluates LLMs via pairwise comparisons (i.e., eliciting preferences for response A vs. B from two models for the same query). Among the $64$ models present in the $57k$ total comparisons, we focus on estimating the win-rate of GPT-4 (specifically, the $11/06$ preview model) vs. Claude 2.1, as they are both strong models, and also have the most pairwise comparisons in the dataset (1073 total), which allows us to get a reliable estimate of $\theta^*$. We model $H$ via the majority preference from 10 Gemini 1.5 Flash (Gemini Team, 2024) evals (5 samples each comparing A vs. B and B vs. A to mitigate position bias). $G$ is the win probability predicted by a Gemma-3 4B model (Gemma Team, 2025) which has been fine-tuned on the other model comparisons from the dataset to predict the Gemini labels. $U$ is computed as $G(1 - G)$. Note that we employ this stylized setup specifically to illustrate how our method can be used to optimize any trade-off between a cheap, weak rater and a more expensive, strong rater (e.g., as opposed to only focusing on the case where the strong rater is a human annotator). With LLM autoevaluation becoming standard practice, the Gemini 1.5 Flash majority vote serves as a realistic, high-cost oracle compared to the computationally cheaper Gemma model. Thus, our focus here is simply in ensuring *consistency* with the expensive metric, but at lower cost.

**Chatbot Arena (estimated easy/hard split).** In an effort to include a dataset with more (identifiable) heteroskedasticy, we also include a filtered version of the GPT-4 versus Claude 2.1 task described above, where we construct a dataset slice containing only the examples corresponding to the bottom 25% and top 25% of Gemma's uncertainty estimates (we use $U$ as the metric). While partly manipulated, this scenario is designed to test for potential gains from actively choosing when to query the expensive rater, as per the intuition from Section 3, where it was shown how higher $\text{Var}(U)$ benefits active policies (though note this may not be true if the estimated $U$ is inaccurate).

**AQA.** Attributed Question Answering (AQA) (Bohnet et al., 2023) assesses if a QA system's answer is both correct *and* supported by the text of a document provided as evidence for it (also by the QA system). We evaluate the highest-scoring "retrieve-and-read" system from the dataset. $H$ is a binary human label that is 1 only if the answer is both correct *and* attributable. $G$ is the probability predicted by an 11B parameter T5 model (Raffel et al., 2020). The T5 model is finetuned on a collection of natural language entailment tasks (Honovich et al., 2022). $U$ is computed as $G(1 - G)$.

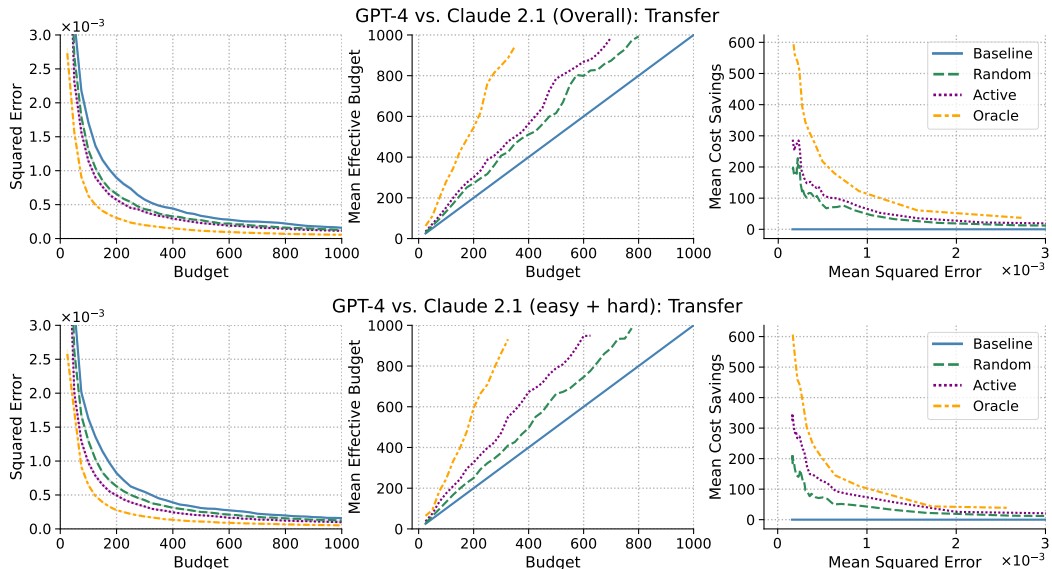

Figure 2: Results on the Chatbot Arena datasets when using policy transfer (see approach A1 in Section 4). Both $\pi_{\text{random}}$ and $\pi_{\text{active}}$ substantially improve estimation quality over $\pi_{\text{base}}$. Consistent with our theory, $\pi_{\text{active}}$'s benefit is magnified on the heterogenous easy/hard split (bottom row).

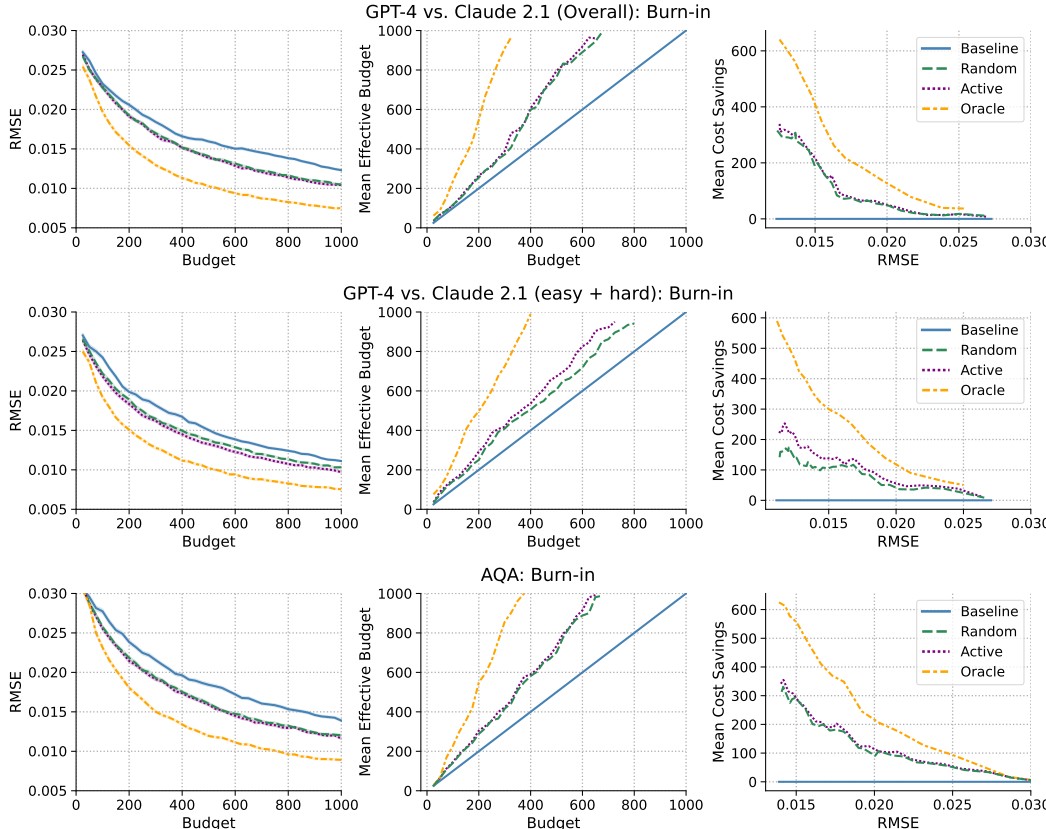

Figure 3: Results on the Chatbot Arena and AQA datasets when using 200 burn-in examples to estimate policy parameters (see approach A2 in Section 4; note that the x-axis reflects the "additional" budget used *after* the burn-in examples). While the absolute differences in RMSE are smaller than in Figure 2, both $\pi_{\text{random}}$ and $\pi_{\text{active}}$ still achieve consistent improvements over $\pi_{\text{base}}$.

### 4.3 RESULTS

Figure 2 shows results for the Chatbot Arena datasets using the *transfer* approach (A1), while Figure 3 shows results for the Chatbot Arena datasets and AQA using the *burn-in* approach (A2).[2] As expected, the absolute improvement for both $\pi_{\text{active}}$ over $\pi_{\text{random}}$ and $\pi_{\text{random}}$ over $\pi_{\text{base}}$ is greatest in the transfer setting in Figure 2, where the parameters of $\pi_{\text{random}}$ and $\pi_{\text{active}}$ can be approximated in advance. In particular, to achieve a root mean-squared error (RMSE) of $0.04$, $\pi_{\text{active}}$ with *estimated parameters* reduces the budget required by $\approx 35\%$ compared to $\pi_{\text{base}}$ in overall setting of Chatbot Arena, and by $\approx 40\%$ in the easy/hard setting. See Figure 6 in the Appendix for relative error and cost reductions. These cost savings become even more pronounced the more precise (i.e., lower MSE) the estimates are required to be. In Figure 3, where the first $n_b = 200$ examples are fully labeled in order to estimate the parameters of $\pi_{\text{random}}$ and $\pi_{\text{active}}$, the absolute *difference* in MSE is smaller for $\pi_{\text{random}}$ and $\pi_{active}$ over $\pi_{\text{base}}$, though the subsequent cost savings over $\pi_{\text{base}}$ for achieving lower and lower MSE (that is, past the MSE of the initial $n_b$ sample estimate) are consistent.

As also predicted by our theory, the results in Figure 2 and Figure 3 show that the extent of the improvement in estimation accuracy varies per dataset. In particular, the best results are obtained on the easy/hard split of the Chatbot Arena dataset, where (i) the weak annotator $G$ is a good proxy of the strong annotator $H$ (both are LLMs), and (ii) there is more variability in the difficulty of examples according to the predicted $U$, resulting in a greater opportunity for improvement for $\pi_{\text{active}}$. On the other hand, while results on AQA and the homogeneous split of the Chatbot Arena dataset also show improvements for $\pi_{\text{random}}$ and $\pi_{\text{active}}$ over $\pi_{\text{base}}$, the relative improvement of $\pi_{\text{active}}$ over $\pi_{\text{random}}$ is fairly small—indicating that while the weak annotator $G$ that is used is relatively good on average, there is not much variability in its estimated uncertainty, $u(x)$, on those distributions. To that point, when we compare to performance using the oracle active policy that uses the true error $u^*(x) = |h(x) - g(x)|^2$ when computing $\pi_{\text{active}}(x)$, it is also clear that the estimated $u(x)$ is also far from perfect. Even on the datasets where the improvement due to the estimated active policy is small, the oracle policy often promises significant headroom: indicating that working on better autorater uncertainty estimation is a promising and important direction for future work in cost-optimal model evaluation.

> **Takeaways: Performance of practical approximations to cost-optimal policies on real data**
>
> Section 2 proved that the optimal policies depend on distributional parameters that must be estimated. How well they are estimated **does not affect the consistency or unbiasedness** of the overall estimator, but it does affect the policy's performance advantage over passive strategies. Yet while estimation is non-trivial, our experiments validate generic recipes that can **successfully approximate the optimal policy**—and yield policies with consistent gains over $\pi_{\text{base}}$.

## 5 CONCLUSION

This paper introduces **theory** and **practice** for **cost-optimal active model evaluations**, a framework that strategically combines cheap raters with more expensive, accurate alternatives to improve evaluation efficiency. Here we theoretically derived annotation policies that are optimal in the sense of minimizing expected error under annotation budget constraints, and we empirically characterized the conditions under which such policies yield improvements over non-hybrid (e.g., human-only) and non-active hybrid alternatives. However, we also showed that the annotation policies that are *optimal in theory* are *distribution dependent*, and therefore include a number of task-specific parameters that are unknown in advance, and must be estimated in practice. Additionally, optimal active annotation depends on having an accurate uncertainty estimates, which can be uncalibrated for AI raters. Nevertheless, we empirically demonstrate that when these parameters are estimated and calibrated on a small burn-in set, we can still achieve strong practical gains over classic evaluation—even while a gap exists to the optimal policy with oracle parameters. Furthermore, many realistic evaluation scenarios involve incrementally adding new models to existing benchmarks; and as shown in §4.3, policy transfer from existing data from other, related evaluations can work quite well. Finally, our results demonstrate that even when active sampling is difficult for the reasons outlined above, the simple, but optimal, fixed sampling rate policy that we derived consistently provides substantial improvements.

---

[2]We also apply power tuning (Angelopoulos et al., 2023b) after all samples are collected. See Appendix B.3.

REPRODUCIBILITY STATEMENT

All proofs of theoretical results are included in Appendix C. Implementation details for all of the empirical experiments are included in Appendix E. All datasets used in for the experiments in Section 4, and the additional experiments in Appendix D, are publicly available. The generation process for the synthetic datasets in Section 3 is described in detail in Sections 3.2 and 3.3.

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

CONTENTS

## A    ETHICS STATEMENT

This paper describes fundamental research on the evaluation of generative AI systems, which is a core technical challenge. Hybrid active evaluation has the potential to improve the cost/accuracy tradeoff of system evaluation, which can make high-quality AI systems easier to build, deploy, and monitor. We do not speculate about broader impacts that may follow from this technical contribution. Gemini was used for light copy-editing during the writing of this work.

## B    ADDITIONAL THEORETICAL RESULTS

### B.1    UNBIASEDNESS OF $\hat{\theta}_T$

For any sequence $\{(H_t, G_t, X_t, \pi_t)\}_{t=1}^T$ where $H_t$ are i.i.d., we have that

$$\mathbb{E}[\hat{\theta}_T] = \mathbb{E}\left[\frac{1}{T}\sum_{t=1}^T \Delta_t\right] = \frac{1}{T}\sum_{t=1}^T \mathbb{E}[G_t] + \mathbb{E}\left[(H_t - G_t)\frac{\xi_t}{\pi_t(X_t)}\right]$$

$$= \frac{1}{T}\sum_{t=1}^T \mathbb{E}[G_t] + \mathbb{E}\left[\mathbb{E}\left[\mathbb{E}\left[(H_t - G_t)\frac{\xi_t}{\pi_t(X_t)} \mid \xi_t\right] \mid \pi_t(X_t)\right]\right]$$

$$= \frac{1}{T}\sum_{t=1}^T \mathbb{E}[G_t] + \mathbb{E}\left[\mathbb{E}\left[H_t - G_t \mid \pi_t(X_t)\right]\right]$$

$$= \frac{1}{T}\sum_{t=1}^T \mathbb{E}[G_t] + \mathbb{E}\left[H_t - G_t\right]$$

$$= \frac{1}{T}\sum_{t=1}^T \mathbb{E}[H_t] = \mathbb{E}[H] = \theta^*$$

### B.2    DERIVATION OF $\mathsf{Error}_T(\pi)$

We provide a short derivation of $\mathsf{Error}_T(\pi)$ in (2). Because the estimator $\hat{\theta}_T^\pi$ is unbiased,

$$\mathbb{E}\left[\left(\hat{\theta}_T^\pi - \theta^*\right)^2\right] = \mathrm{Var}(\hat{\theta}_T^\pi) = \frac{1}{T}\mathrm{Var}(\Delta^\pi)$$

when $\pi$ and $g$ are fixed, and where $\Delta^\pi = G + (H - G)\frac{\xi}{\pi(X)}$. Then,

$$\mathrm{Var}(\Delta^\pi) = \mathbb{E}\left[\left(G + (H - G)\frac{\xi}{\pi(X)}\right)^2\right] - (\theta^*)^2$$

$$= \mathbb{E}\left[G^2\right] + \mathbb{E}\left[\left((H - G)\frac{\xi}{\pi(X)}\right)^2\right] + 2\mathbb{E}\left[G(H - G)\frac{\xi}{\pi(X)}\right] - (\theta^*)^2$$

$$= \mathbb{E}\left[G^2\right] + \mathbb{E}\left[(H - G)^2\frac{1}{\pi(X)}\right] + 2\mathbb{E}\left[G(H - G)\right] - (\theta^*)^2$$

$$= -(\theta^*)^2 + 2\mathbb{E}[GH] - \mathbb{E}[G^2] + \mathbb{E}\left[(H - G)^2\frac{1}{\pi(X)}\right]$$

$$= \mathbb{E}[H^2] - (\theta^*)^2 - \mathbb{E}[H^2] + 2\mathbb{E}[GH] - \mathbb{E}[G^2] + \mathbb{E}\left[(H - G)^2\frac{1}{\pi(X)}\right]$$

$$= \mathrm{Var}(H) - \mathbb{E}[(H - G)^2] + \mathbb{E}\left[(H - G)^2\frac{1}{\pi(X)}\right].$$

### B.3 POWER TUNING

Angelopoulos et al. (2023b) proposed "power tuning" as a way to improve upon the standard PPI estimator by allowing the estimator to adapt to the "usefulness" of the supplementary predictions (here, the weak rater G) with a tuning parameter $\lambda \in \mathbb{R}$. We now extend this to our setting.

Let us consider a modified version of our estimator, with some fixed policy $\pi$ and $\lambda \in \mathbb{R}$:

$$\hat{\theta}_T^\lambda = \frac{1}{T} \sum_{t=1}^{T} \lambda G_t + (H_t - \lambda G_t) \frac{\xi_t}{\pi(X_t)}.$$

For all values of $\lambda$, this estimator is unbiased. Our job is to pick the value with minimum error. Following the previous derivation in Section B.2, the error of the estimator is

$$\mathsf{Error}_{T,\pi}(\lambda) = \frac{1}{T} \left( \mathrm{Var}(H) - \mathbb{E}[(H - \lambda G)^2] + \mathbb{E}\left[ (H - \lambda G)^2 \frac{1}{\pi(X)} \right] \right),$$

which is optimized by

$$\lambda^* = \operatorname*{argmin}_{\lambda \in \mathbb{R}} \mathbb{E}\left[ (H - \lambda G)^2 \left( \frac{1}{\pi(X)} - 1 \right) \right]$$

$$= \operatorname*{argmin}_{\lambda \in \mathbb{R}} \lambda^2 \mathbb{E}\left[ G^2 \left( \frac{1}{\pi(X)} - 1 \right) \right] - 2\lambda \mathbb{E}\left[ HG \left( \frac{1}{\pi(X)} - 1 \right) \right].$$

The above expression is quadratic in $\lambda$, and its optimizer is

$$\lambda^* = \frac{\mathbb{E}\left[ HG \left( \frac{1}{\pi(X)} - 1 \right) \right]}{\mathbb{E}\left[ G^2 \left( \frac{1}{\pi(X)} - 1 \right) \right]},$$

which can be estimated in any consistent way, e.g., by its prediction-powered plug-in that can be computed after sampling all $(X_t, G_t, H_t, \xi_t)$ as:

$$\hat{\lambda}_T = \frac{\frac{1}{T} \sum_{t=1}^{T} \left( G_t^2 + (H_t G_t - G_t^2) \frac{\xi_t}{\pi_t(X_t)} \right) \left( \frac{1}{\pi_t(X_t)} - 1 \right)}{\frac{1}{T} \sum_{t=1}^{T} G_t^2 \left( \frac{1}{\pi_t(X_t)} - 1 \right)}.$$

### B.4 OPTIMAL RANDOM ANNOTATION: DISCRETE TIME CASE

The following proposition is the full version of Proposition 1—with the constraint that $T^{\mathrm{stop}}$ is an integer. This leads to a substantially more complex optimization problem; we show the solution here, but we do not implement it in practice.

**Proposition 4.** *Let* $(X_1, G_1, H_1), \ldots, (X_T, G_T, H_T)$, $T \in \mathbb{N}$, *be an i.i.d. sequence of real-valued random variables with joint distribution $P$, and define* Error, Cost, *and* $\Pi^{\mathrm{random}}$ *as above. Additionally, define the optimization problem*

$$\begin{aligned} \operatorname*{minimize}_{\substack{\pi \in \Pi^{\mathrm{random}} \\ T^{\mathrm{stop}} \in \mathbb{N}_+}} \quad & \mathsf{Error}_{T^{\mathrm{stop}}}(\pi) \\ \text{subject to} \quad & \mathsf{Cost}_{T^{\mathrm{stop}}}(\pi) \leq B. \end{aligned} \tag{7}$$

*Then the optimal solution to Problem (7) is either $\pi^*(x) = 1$ or*

$$\pi^*(x) = \frac{B - k^* c_g}{k^* c_h}.$$

*for all $x \in \mathcal{X}$, where*

$$k^* = \operatorname*{argmin}_{k \in \mathcal{K}} \frac{1}{k} \left( \mathrm{Var}(H) - \mathbb{E}[(H - G)^2] \right) + \frac{c_h}{B - k c_g} \mathbb{E}[(H - G)^2],$$

*and*

$$\mathcal{K} = \left\{ \left\lfloor B \frac{1 + \sqrt{\frac{c_h}{c_g} \frac{\mathbb{E}[(H-G)^2]}{\mathrm{Var}(H) - \mathbb{E}[(H-G)^2]}}}{c_g - c_h \frac{\mathbb{E}[(H-G)^2]}{\mathrm{Var}(H) - \mathbb{E}[(H-G)^2]}} \right\rfloor, \left\lceil B \frac{1 + \sqrt{\frac{c_h}{c_g} \frac{\mathbb{E}[(H-G)^2]}{\mathrm{Var}(H) - \mathbb{E}[(H-G)^2]}}}{c_g - c_h \frac{\mathbb{E}[(H-G)^2]}{\mathrm{Var}(H) - \mathbb{E}[(H-G)^2]}} \right\rceil \right\}.$$

It is easy to disambiguate between $p^* = 1$ and the optimal policy based on $k^*$ by comparing the objective values directly.

### B.5 EXTENSION TO CONVEX M-ESTIMATORS

Here we give an extension of Proposition 2 to general convex M-estimators (Van der Vaart, 2000). Consider a convex loss function, $\ell_\theta$ for some $\theta \in \mathbb{R}^d$, equipped with the simplified notation $\ell_{\theta,t} = \ell_\theta(X_t, H_t)$ for all $t \in \mathbb{N}$ and $\ell_{\theta,t}^g = \ell_\theta(X_t, G_T)$. We also use $\ell_\theta = \ell_\theta(X, H)$ and $\ell_\theta^g = \ell_\theta(X, G)$ for generic points $(X, G, H) \sim P$. The target of estimation is the population minimizer, $\theta^* = \operatorname{argmin}_{\theta \in \mathbb{R}^d} \mathbb{E}[\ell_\theta]$. The active estimator is

$$\hat{\theta}_T = \operatorname*{argmin}_{\theta \in \mathbb{R}^d} \frac{1}{T} \sum_{t=1}^{T} \Delta_{\theta,t} \quad \text{where } \Delta_{\theta,t} = \ell_{\theta,t}^g + \left( \ell_{\theta,t} - \ell_{\theta,t}^g \right) \frac{\xi_t}{\pi_t(X_t)},$$

for some sequence of annotation policies $\pi_t$, $t \in \mathbb{N}$. For the purpose of deriving optimal annotation policies when $\pi_t$ is fixed as in Section 2, we will also define

$$\hat{\theta}_T^\pi = \operatorname*{argmin}_{\theta \in \mathbb{R}^d} \frac{1}{T} \sum_{t=1}^{T} \Delta_{\theta,t} \quad \text{where } \Delta_{\theta,t} = \ell_{\theta,t}^g + \left( \ell_{\theta,t} - \ell_{\theta,t}^g \right) \frac{\xi_t}{\pi(X_t)}.$$

Unlike the estimator in the case of mean estimation from Section 2, $\hat{\theta}_T^\pi$ does not have a closed-form variance in finite samples. The standard solution in the analysis of M-estimators is to appeal to the asymptotic linearity of M-estimators to analyze the variance (Van der Vaart, 2000), as is done in Theorem 1 of Zrnic & Candès (2024). The result below combines the aforementioned theorem with standard parametric analysis to give the asymptotic distribution of the squared error.

**Proposition 5.** *Let $\ell_\theta$ be smooth (see Assumption 1 in (Zrnic & Candès, 2024)) and define the Hessian $W_{\theta^*} = \nabla^2 \mathbb{E}[\ell_{\theta^*,t}]$. Then if $\hat{\theta}_T^\pi \xrightarrow{p} \theta^*$, we have*

$$\sqrt{T}(\hat{\theta}_T^\pi - \theta^*) \xrightarrow{d} \mathcal{N}(0, \Sigma^*),$$

*where $\Sigma^* = W_{\theta^*}^{-1} \operatorname{Var}\left( \nabla \ell_{\theta^*,t}^g + \left( \nabla \ell_{\theta^*,t} - \nabla \ell_{\theta^*,t}^g \right) \frac{\xi_t}{\pi(X_t)} \right) W_{\theta^*}^{-1}$. Therefore, we have*

$$T \left\| \hat{\theta}_T^\pi - \theta^* \right\|_2^2 \xrightarrow{d} \sum_{j \in [d]} \lambda_j \zeta_j,$$

*where $\zeta_j \overset{\text{i.i.d.}}{\sim} \chi_1^2$ for all $j \in [d]$ and $\lambda_j$ is the $j$th eigenvalue of $\Sigma^*$.*

The above proposition gives us consistency of the active estimator, and more importantly, the asymptotic distribution of the squared error. Since $\mathbb{E}[\zeta_j] = 1$ for all $j$, and the sum of the eigenvalues of a square matrix is equal to the trace, we know the mean-squared error is asymptotically equal to $\operatorname{Error}_T(\pi) = \frac{1}{T} \operatorname{Tr}(\Sigma^*)$. With this in hand, we can use the same strategy from earlier to find the optimal annotation policy, using the asymptotic approximation of the error. For simplicity, here we assume that we are always on the interior of the constrained optimization problem, i.e., we solve for unconstrained $\pi(x)$ while assuming that $\gamma^* \sqrt{u(X)} \leq 1$. That said, a more rigorous treatment analogous to that in Proposition 2 can also be applied here, which we leave to future work.

**Proposition 6.** *In the setting of Proposition 5, let $(X_1, G_1, H_1), \ldots, (X_T, G_T, H_T)$, $T \in \mathbb{N}$, be an i.i.d. sequence of real-valued random variables with joint distribution $P$, and define $\operatorname{Error}_T(\pi) = \frac{1}{T} \sum_{j \in [d]} \operatorname{Tr}(\Sigma^*)$. Furthermore, define $\operatorname{Cost}$ and $\Pi$ as in Proposition 2.*

*Construct the optimization problem*

$$\begin{aligned}
\underset{\pi \in \mathcal{F},\ T^{\text{stop}} \in \mathbb{R}_{>0}}{\text{minimize}} \quad & \operatorname{Error}_{T^{\text{stop}}}(\pi) \\
\text{subject to} \quad & \operatorname{Cost}_{T^{\text{stop}}}(\pi) \leq B.
\end{aligned} \tag{8}$$

*where $\mathcal{F} = \{x \mapsto f(x) : f(x) \in (0, \infty); \forall x \in \mathcal{X}\}$. Then the solution to Problem (8) is*

$$\pi^*(x) = \sqrt{\frac{c_g}{c_h} \cdot \frac{u(x)}{C}}$$

*where*

$$u(x) = \mathbb{E}\left[\mathrm{Tr}\left(W_{\theta^*}^{-1}\left(\nabla\ell_{\theta^*} - \nabla\ell_{\theta^*}^g\right)\left(\nabla\ell_{\theta^*} - \nabla\ell_{\theta^*}^g\right)^\top W_{\theta^*}^{-1}\right) \mid X = x\right],$$

*and*

$$C = \mathrm{Tr}\left(W_{\theta^*}^{-1}\left(\mathbb{E}\left[\nabla\ell_{\theta^*}^g(\nabla\ell_{\theta^*})^\top + (\nabla\ell_{\theta^*} - \nabla\ell_{\theta^*}^g)(\nabla\ell_{\theta^*}^g)^\top\right] - \mathbb{E}[\nabla\ell_{\theta^*}]\mathbb{E}[\nabla\ell_{\theta^*}]^\top\right)W_{\theta^*}^{-1}\right).$$

**Remark 7.** *When $\pi^*(x) \leq 1$, $\forall x \in \mathcal{X}$, then $\pi^*$ is also optimal for Problem (8) solved for $\pi \in \Pi$.*

### B.5.1    MEAN ESTIMATION

In the case of mean estimation, the loss function takes the form

$$\ell_\theta(x, h) = \frac{1}{2}(h - \theta)^2,$$

where $\nabla\ell_{\theta^*}(X, H) - \nabla\ell_{\theta^*}(X, G) = H - G$, and $W_{\theta^*}$ is the identity matrix. Plugging back into $\pi^*$ in Proposition 6 recovers $\pi_{\mathrm{active}}$ from Proposition 2 without clipping ($\tau^* = \infty$), i.e.,

$$\sqrt{\frac{c_g}{c_h}\frac{\mathbb{E}[(H - G)^2 \mid X = x]}{\mathrm{Var}(H) - \mathbb{E}[(H - G)^2]}}.$$

### B.5.2    GENERALIZED LINEAR MODELS

In the case of GLMs, the loss function takes the form

$$\ell_\theta(x, h) = -hx^\top\theta + \psi(x^\top\theta)$$

for some convex log-partition function $\psi$. Thus, $\nabla\ell_{\theta^*}(X, H) - \nabla\ell_{\theta^*}(X, G) = (G - H)X$. So, again by the linearity of the trace, we have that

$$\pi^*(x) \propto \sqrt{\mathbb{E}\left[(H - G)^2 \mid X = x\right]\mathrm{Tr}\left(W_{\theta^*}^{-1}xx^\top W_{\theta^*}^{-1}\right)}.$$

### B.6    EFFECTS OF NOISY POLICY PARAMETERS ON ESTIMATOR VARIANCE

In practice, we will be using only an imperfect estimate of $u(x)$ for $\pi_{\mathrm{active}}$, which can negatively affect the performance of $\pi_{\mathrm{active}}$ to a substantial degree, as we have seen for some of the datasets in Section 4. Similarly, we will also only be using imperfect estimates of the optimal scaling and thresholding parameters used in $\pi_{\mathrm{active}}$, which further limit performance.

There are two main factors that affect the error of a policy:

1. The variance, $\mathrm{Var}(\Delta^\pi)$, of each active increment $\Delta^\pi$, where $\Delta^\pi = G + (H - G)\frac{\xi}{\pi(X)}$.

2. The average sample size at which the estimator runs out of budget, $T^{\mathrm{stop}} = \frac{B}{c_h\mathbb{E}[\pi(X)] + c_g}$.

In this section, we provide some additional theoretical analysis on the first factor, i.e., the increase in $\mathrm{Var}(\Delta^\pi)$ due to the mispecification error of an *estimated* active policy, while noting that the total error will be further affected by the relative increase/decrease of the mean sampling rate, $\mathbb{E}[\pi(X)]$.

**Proposition 8.** *In the same setting as Proposition 2, let $\tilde{\pi} : \mathcal{X} \to (0, 1]$ be any function satisfying*

$$\mathbb{E}\left[\frac{1}{\tilde{\pi}(X)} - \frac{1}{\pi^*(x)}\right] \leq \delta,$$

*where $\pi^*$ is the oracle estimate of $\pi_{\mathrm{active}}$. Let $(H - G)^2 \overset{\mathrm{a.s.}}{\leq} b$. Then $\mathrm{Var}(\Delta^{\tilde{\pi}}) \leq \mathrm{Var}(\Delta^{\pi^*}) + b\delta$.*

If we simply things by assuming an additive error model for a policy without thresholding (i.e., $\tau^* = \infty$), we can refine the bound somewhat further:

**Corollary 9.** *Let $\tilde{\pi} = \tilde{\gamma}\sqrt{\tilde{u}(x)}$, where $\tilde{\gamma} = \gamma^* + \delta_\gamma$, $\tilde{u}(x) = u(x) + \delta_u(x)$, and $u(X) \overset{a.s.}{\geq} \epsilon$. Further assume that $\tilde{\pi}$ is admissible, i.e., $\tilde{\pi}(x) \in (0,1] \; \forall x$. Then, up to first-order terms in $\delta_\gamma$ and $\delta_u(x)$,*

$$\text{Var}(\Delta^{\tilde{\pi}}) \leq \text{Var}(\Delta^{\pi^*}) + b\left(\frac{|\delta_\gamma|}{(\gamma^*)^2\sqrt{\epsilon}} + \frac{1}{2\gamma^*\epsilon^{3/2}}\mathbb{E}[|\delta_U(X)|]\right).$$

We can make a few observations about the results in Proposition 8 and Corollary 9. First, as long as the error, $(H-G)^2$ is bounded, and the estimated inverse propensity score $1/\tilde{\pi}(X)$ is not significantly higher than the oracle inverse propensity score $1/\pi^*(X)$ on average, then the increase in variance over the oracle will not be that large. Generally speaking, this is satisfied when the estimated policy is not *overconfident* on examples that in fact have high error. Of course, regularizing the estimated policy to be underconfident on all examples is also not always a satisfying solution: as $\mathbb{E}[\tilde{\pi}(X)] \to 1$, we obtain a policy that is no better than $\pi_{\text{base}}$. Similarly, as seen in Section 3, the headroom for $\pi_{\text{active}}$ over $\pi_{\text{base}}$ is largest when $(H-G)^2$ is *not* bounded (e.g., the Gaussian data setting compared to the Bernoulli data setting), as large $(H-G)^2$ also increase the possible variance of $U$. This reinforces the importance of having **accurate uncertainty estimates** when computing active policies.

### B.7 OPTIMAL ACTIVE SAMPLING OF INPUT EVALUATION QUERIES

This section shows how to optimally choose the distribution of $X$. In contrast, Section 2 in the main paper focuses only on querying annotators for $H$ given i.i.d. samples from the fixed distribution $P$ for $X$. Deciding *which inputs* to sample can be a more difficult problem than deciding whether to annotate a given input sample because $\mathcal{X}$ can be large and complex. However, we can always apply the optimal rules to a coarse stratification of $\mathcal{X}$. Towards this end, we define the estimator

$$\hat{\theta}_T^{Q,\pi} = \frac{1}{T}\sum_{t=1}^{T}\Delta_t, \text{ where } \quad \Delta_t^{Q,\pi} = \frac{dP}{dQ}(X_t)\left(G_t + (H_t - G_t)\frac{\xi_t}{\pi(X_t)}\right),$$

$$X_t \overset{\text{i.i.d.}}{\sim} Q, \; H_t \sim P_{H|X}, \; G_t \sim P_{G|X}$$

which is our previous estimator with a fixed policy $\pi$, and where the $X$ are sampled from a distribution $Q$, and the distribution of $H \mid X$ and $G \mid X$ remain unchanged. This estimator is unbiased for $\theta^*$, and a straightforward calculation gives that the error of the estimator is

$$\text{Error}_T(Q,\pi) = \mathbb{E}_Q\left[\left(\hat{\theta}_T^{Q,\pi} - \theta^*\right)^2\right]$$

$$= \frac{1}{T}\text{Var}(\Delta^{Q,\pi})$$

$$= \frac{1}{T}\left(\mathbb{E}_P\left[\frac{dP}{dQ}(X)\left(H^2 + \left(\frac{1}{\pi(X)} - 1\right)(H-G)^2\right)\right] - (\theta^*)^2\right).$$

The goal is to pick a distribution $Q$ to minimize the error of the estimator. The following proposition gives an explicit form for this optimal sampling distribution.

**Proposition 10.** *Define $\text{Error}_T$ as above, and define the set of all strictly positive densities, $\mathcal{Q} = \{x \mapsto Q(x) : Q(x) \in \mathbb{R}_{>0} \text{ and } Q \in \Delta^{\mathcal{X}}\}$. Furthermore, define the optimization problem*

$$\underset{Q \in \mathcal{Q}}{\text{minimize}} \quad \text{Error}_T(Q,\pi) \tag{9}$$

*for a fixed time $T \in \mathbb{N}$. Then the solution to Problem (3) is*

$$Q^*(x) = \mathbb{P}(X = x)\frac{\sqrt{\nu(x)}}{\mathbb{E}_P[\sqrt{\nu(X)}]},$$

*where*

$$\nu(x) = \mathbb{E}_P\left[\left(H^2 + \left(\frac{1}{\pi(X)} - 1\right)(H-G)^2\right) \mid X = x\right]$$

*for all $x \in \mathcal{X}$.*

We leave empirical exploration of active input sampling to future work.

## B.8 INFORMATIVE SPECIAL CASES FOR $\pi_{\text{active}}$

Prior work (Zrnic & Candès, 2024; Gligorić et al., 2024) target some fixed, prespecified value (i.e., some ratio $n/N$ for $\mathbb{E}[\pi(X)]$). A key distinction of this work is that we optimize $\mathbb{E}[\pi(X)]$, which will depend strongly on $c_g/c_h$, that is, the cost ratio of $G$ to $H$. In this section we analyze two extreme, but informative cases, for active sampling when either $c_g/c_h = 0$ or $c_g/c_h = \infty$, that serve to illustrate how $\mathbb{E}[\pi_{\text{active}}(X)]$ for the cost-optimal policy $\pi_{\text{active}}$ can consequently be as extreme as 0 or 1.

OPTIMAL POLICY FOR $c_g = 0$

We start with the special case where $c_g = 0$, so that we can obtain essentially infinitely many queries of the weak rater $G$ irrespective of the budget constraint. In this case, we expect that unless $G$ has a prohibitively large error $\mathbb{E}[(H - G)^2]$, we can purely rely on querying $G$, and overcome its error with sufficiently many samples. Indeed, let us assume that $\mathbb{E}[(H - G)^2] = \mathbb{E}[U] < \text{Var}(H)$. Then we note that for any $\tau > 0$:

$$\tau\sqrt{\frac{c_g/c_h + \mathbb{P}(U > \tau^2)}{\text{Var}(H) - \mathbb{E}[U\mathbb{1}\{U \leq \tau^2\}]}} = \sqrt{\frac{\tau^2\mathbb{P}(U > \tau^2)}{\text{Var}(H) - \mathbb{E}[U\mathbb{1}\{U \leq \tau^2\}]}}$$

$$\leq \sqrt{\frac{\tau^2\mathbb{P}(U > \tau^2)}{\mathbb{E}[U] - \mathbb{E}[U\mathbb{1}\{U \leq \tau^2\}]}}$$

$$= \sqrt{\frac{\tau^2\mathbb{P}(U > \tau^2)}{\mathbb{E}[U\mathbb{1}\{U > \tau^2\}]}} \leq 1,$$

where the first inequality is due to our assumption that $\mathbb{E}[U] < \text{Var}(H)$, and the last inequality follows from $\mathbb{E}[U\mathbb{1}\{U > \tau^2\}] > \tau^2\mathbb{P}(U > \tau^2)$. Consequently, we get that in this case,

$$\gamma^*(\tau) = \sqrt{\frac{\mathbb{P}(U > \tau^2)}{\text{Var}(H) - \mathbb{E}[U\mathbb{1}\{U \leq \tau^2\}]}}.$$

Suppose for now that we only consider the values $\tau$ where $U$ further satisfies that $\sqrt{U}\gamma^*(\tau) \leq 1$ almost surely, and denote this set by $\mathcal{T}$. Let $\Delta = \text{Var}(H) - \mathbb{E}[U] > 0$. Then we see that

$$\min_{\tau \in \mathcal{T}} c_h\mathbb{E}[\pi_{\text{clip}}(x; \tau)]\left[\Delta + \mathbb{E}\left[\frac{U}{\pi_{\text{clip}}(x; \tau)}\right]\right]$$

$$= \min_{\tau \in \mathcal{T}} c_h\mathbb{E}[\sqrt{U}\gamma^*(\tau)]\left[\Delta + \mathbb{E}\left[\frac{U}{\sqrt{U}\gamma^*(\tau)}\right]\right]$$

$$= \min_{\tau \in \mathcal{T}} c_h\mathbb{E}[\sqrt{U}\gamma^*(\tau)]\Delta + c_h\mathbb{E}[\sqrt{U}\gamma^*(\tau)]\mathbb{E}\left[\frac{\sqrt{U}}{\gamma^*(\tau)}\right].$$

Since $\gamma^*(\tau)$ is deterministic, it cancels from the second term above, and we get that the annotation cost over $\tau \in \mathcal{T}$ is monotonically increasing in $\gamma^*(\tau)$, meaning that we choose $\tau \to \infty$, which yields $\gamma^*(\tau) \to 0$ (since $P(U > \infty) = 0$). We also note that whenever $\sqrt{U} \leq B$, all $\tau$ such that $\gamma^*(\tau) < 1/B$ are in $\mathcal{T}$ trivially, since this satisfies $\sqrt{U}\gamma^*(\tau) < 1$. In particular, this includes our choice of $\tau \to \infty$, which ensures that $\gamma^*(\tau) \to 0$. Finally, we note that from the proof of Proposition 2 (specifically, Equation 12), we have that $\pi(x) = \gamma\sqrt{u(x)}$ minimizes the objective

$$(c_hE[\pi(X)] + c_g)\left[\text{Var}(H) - \mathbb{E}[U] + \mathbb{E}\left[\frac{U}{\pi(X)}\right]\right],$$

over all mappings $\pi \in \{x \mapsto f(x) : f(x) \in (0, \infty); \forall x \in \mathcal{X}\}$. Since we find that our optimal choice without imposing the constraint $\pi(x; \tau) \leq 1$ is already feasible, it is also optimal for the constrained problem, $\pi \in \{x \mapsto f(x) : f(x) \in (0, 1]; \forall x \in \mathcal{X}\}$.

OPTIMAL POLICY FOR $c_h = 0$

The other extreme case is simpler. When $c_h = 0$, the objective for $\tau^*$ becomes monotonically decreasing in $\pi(x; \tau)$. If we assume that $\tau$ is such that $\gamma^*(\tau) < 1/\tau$, then we find that the expression

$$\sqrt{\frac{c_g/c_h + \mathbb{P}(U > \tau^2)}{(\mathrm{Var}(H) - \mathbb{E}[U\mathbb{1}\{U \le \tau^2\}])_+}}$$

becomes infinite due to $c_h = 0$, and hence we must have $\gamma^*(\tau) = 1/\tau$. However, for any $x$ such that $\pi(x; \tau) < 1$, we have $1/\pi(x; \tau) = \tau/\sqrt{u(x)}$. Consequently, minimizing over $\tau$ results in $\tau = 0$. But this gives $\pi(x; \tau) = \infty$, so that we must have $\pi(x; \tau) = 1$ for all $x$. Intuitively, this makes sense because any $\pi(x; \tau) < 1$ results in an estimator with variance strictly greater than $\mathrm{Var}(H)$, but having $\pi(x; \tau) \equiv 1$ allows us to attain the smallest possible variance of $\mathrm{Var}(H)$. Since there is no effect of these choices on the estimation cost, we choose the lowest variance estimator in this case, and direct all our queries to the strong rater.

## C PROOFS

### C.1 PROOF OF PROPOSITION 1

*Proof.* Since $\mathrm{Error}_T(\pi)$ is monotone in $T$ for all $\pi$, we should first set $T^{\mathrm{stop}}$ to be the largest $T$ for which the constraint holds. This value is

$$T^{\mathrm{stop}} = \frac{B}{c_h p + c_g}.$$

Plugging this into the objective yields

$$(c_h p + c_g)\left(\mathrm{Var}(H) - \mathbb{E}[(H - G)^2] + \frac{1}{p}\mathbb{E}\left[(H - G)^2\right]\right),$$

which, after removing terms that do not depend on $p$, is equivalent to minimizing

$$c_h p \left(\mathrm{Var}(H) - \mathbb{E}[(H - G)^2]\right) + \frac{c_g}{p}\mathbb{E}\left[(H - G)^2\right]$$

subject to the constraint that $p \in [0, 1]$.

This is a convex problem in $p$, and we know that the solution lies either on the boundary or on the interior. We will compare the values of the objectives in three cases: $p^* = 0$, $p^* = 1$, and $p^* \in (0, 1)$. It is clear that $p^* = 0$ is infeasible (unless $H \stackrel{\mathrm{a.s.}}{=} G$, which renders the problem trivial) because the factor $c_g/p$ appears in the above objective. In the case that $p^* = 1$, the objective value is

$$c_h \left(\mathrm{Var}(H) - \mathbb{E}[(H - G)^2]\right) + c_g \mathbb{E}\left[(H - G)^2\right].$$

In the case that $p^* \in (0, 1)$, it must be a critical value, so it satisfies the first-order condition

$$c_h \left(\mathrm{Var}(H) - \mathbb{E}[(H - G)^2]\right) = \frac{c_g}{(p^*)^2}\mathbb{E}[(H - G)^2],$$

and thus,

$$(p^*)^2 = \frac{c_g \mathbb{E}[(H - G)^2]}{c_h(\mathrm{Var}(H) - \mathbb{E}[(H - G)^2])}. \tag{10}$$

However, because we are in the case $p^* \in (0, 1)$, the right-hand side above must be positive (otherwise the square root would be imaginary), and it cannot be greater than 1 (otherwise we would have $p^* > 1$, which is a contradiction). This gives us that

$$p^* \in (0, 1) \implies \mathbb{E}[(H - G)^2] < \mathrm{Var}(H) \text{ and } (c_g + c_h)\mathbb{E}[(H - G)^2] < c_h\mathrm{Var}(H).$$

Under these conditions, we can take square roots on both sides of (10) to obtain

$$p^* = \sqrt{\frac{c_g}{c_h}\frac{1}{\frac{\mathrm{Var}(H)}{\mathbb{E}[(H-G)^2]} - 1}}.$$

The objective value at this point is

$$2\sqrt{c_g c_h}\sqrt{\mathbb{E}[(H-G)^2](\operatorname{Var}(H) - \mathbb{E}[(H-G)^2])}.$$

Finally, comparing the above objective value with that of $p^* = 1$, we have that

$$2\sqrt{c_g c_h}\sqrt{\mathbb{E}[(H-G)^2](\operatorname{Var}(H) - \mathbb{E}[(H-G)^2])} < c_h\left(\operatorname{Var}(H) - \mathbb{E}[(H-G)^2]\right) + c_g\mathbb{E}\left[(H-G)^2\right]$$

$$\Longleftrightarrow 0 < c_h^2\left(\operatorname{Var}(H) - \mathbb{E}[(H-G)^2]\right)^2 - 2c_g c_h\mathbb{E}[(H-G)^2](\operatorname{Var}(H) - \mathbb{E}[(H-G)^2]) + c_g^2\mathbb{E}\left[(H-G)^2\right]^2$$

$$\Longleftrightarrow 0 < \left(c_h(\operatorname{Var}(H) - \mathbb{E}[(H-G)^2]) - c_g\mathbb{E}[(H-G)^2]\right)^2.$$

Under the condition that $(c_g + c_h)\mathbb{E}[(H-G)^2] < c_h\operatorname{Var}(H)$, the above inequality cannot hold, since the squared term on the right-hand side will always be positive (and nonzero). Thus, we have that

$$p^* = \begin{cases} \sqrt{\frac{c_g}{c_h}\frac{1}{\frac{\operatorname{Var}(H)}{\mathbb{E}[(H-G)^2]} - 1}} & \text{if } (c_g + c_h)\mathbb{E}[(H-G)^2] < c_h\operatorname{Var}(H) \text{ and } \mathbb{E}[(H-G)^2] < \operatorname{Var}(H) \\ 1 & \text{otherwise.} \end{cases}$$

Under the constraint that $c_h \geq c_g$, this simplifies to

$$p^* = \begin{cases} \sqrt{\frac{c_g}{c_h}\frac{\mathbb{E}[(H-G)^2]}{\operatorname{Var}(H) - \mathbb{E}[(H-G)^2]}} & \text{if } \mathbb{E}[(H-G)^2] < \frac{c_h}{c_g + c_h}\operatorname{Var}(H) \\ 1 & \text{otherwise.} \end{cases}$$

$\square$

## C.2 PROOF OF PROPOSITION 2

*Proof.* Following the simplification of Problem (3) in the proof of Proposition (1), Problem (5) is also equivalent to minimizing the following objective:

$$J(\pi) = c_h\mathbb{E}[\pi(X)]\left(\operatorname{Var}(H) - \mathbb{E}[(H-G)^2] + \mathbb{E}\left[(H-G)^2\frac{1}{\pi(X)}\right]\right) + c_g\mathbb{E}\left[(H-G)^2\frac{1}{\pi(X)}\right].$$

At this point, we leverage the discreteness of $\mathcal{X}$ to write the objective in a simpler form. Let $P \in \Delta^{\mathcal{X}}$ be the probability mass function of $X$, expressed as a vector, and let $I \in \{0,1\}^{|\mathcal{X}|}$ be the indicator that $X$ takes each value in $\mathcal{X}$. Furthermore, let $p \in [0,1]^{|\mathcal{X}|}$ be the vector of $\pi(x)$ for all $x \in \mathcal{X}$. Then, we can express $\pi(X) = p^\top I$ and $\mathbb{E}[\pi(X)] = p^\top P$, and write the objective as

$$J(\pi) = J(p) = p^\top P\left(\operatorname{Var}(H) - \mathbb{E}[(H-G)^2] + \mathbb{E}\left[(H-G)^2\frac{1}{p^\top I}\right]\right) \tag{11}$$

$$+ \frac{c_g}{c_h}\mathbb{E}\left[(H-G)^2\frac{1}{p^\top I}\right].$$

From here on out, we assume that $P_x > 0$ for all $x \in \mathcal{X}$. The final result will hold without loss of generality, since the value of the optimal policy on measure-zero points does not change the value of the objective. For any $x$, we clearly cannot have $p_x = 0$, otherwise the objective would be infinite. This rules out $p_x = 0$ for almost all $x$. We are left with the constraint that $p \preceq 1$.

Forming the Lagrangian,

$$\mathcal{L}(p, \lambda) = J(p) + \lambda^\top(p - 1)$$

$$= p^\top P\left(\operatorname{Var}(H) - \mathbb{E}[(H-G)^2] + \mathbb{E}\left[(H-G)^2\frac{1}{p^\top I}\right]\right) + \frac{c_g}{c_h}\mathbb{E}\left[(H-G)^2\frac{1}{p^\top I}\right] + \lambda^\top(p-1).$$

Taking the gradient with respect to $p$ gives $\nabla_p\mathcal{L}(p, \lambda)$ equal to

$$P\left(\operatorname{Var}(H) - \mathbb{E}[(H-G)^2]\right) - \left(p^\top P + \frac{c_g}{c_h}\right)\mathbb{E}\left[(H-G)^2\frac{I}{(p^\top I)^2}\right] + P\mathbb{E}\left[(H-G)^2\frac{1}{p^\top I}\right] + \lambda.$$

Setting the gradient to zero coordinate-wise then gives that for each $x$,

$$P_x \left( \text{Var}(H) - \mathbb{E}[(H-G)^2] \mathbb{E}\left[ (H-G)^2 \frac{1}{p^\top I} \right] \right) = \left( p^\top P + \frac{c_g}{c_h} \right) \mathbb{E}\left[ (H-G)^2 \frac{I_x}{p_x^2} \right] - \lambda_x.$$

By the definition of the conditional expectation, and rearranging, we can rewrite this as

$$\text{Var}(H) - \mathbb{E}[(H-G)^2] + \mathbb{E}\left[ (H-G)^2 \frac{1}{p^\top I} \right] + \frac{\lambda_x}{P_x} = \left( p^\top P + \frac{c_g}{c_h} \right) \mathbb{E}\left[ (H-G)^2 \frac{1}{p_x^2} \mid X = x \right].$$

Solving for the optimal value as a function of the Lagrange multipliers $\lambda$ gives the following expression:

$$p_x(\lambda)^2 = \frac{\left( p^\top P + \frac{c_g}{c_h} \right) \mathbb{E}\left[ (H-G)^2 \mid X = x \right]}{\text{Var}(H) - \mathbb{E}[(H-G)^2] + \mathbb{E}\left[ (H-G)^2 \frac{1}{p^\top I} \right] + \frac{\lambda_x}{P_x}}.$$

The denominator of this expression is always positive, since for all valid $p$, $\frac{(H-G)^2}{p^\top I} \overset{\text{a.s.}}{\geq} (H-G)^2$, and the remaining terms are positive. Thus,

$$p_x(\lambda) = \sqrt{\frac{\left( p^\top P + \frac{c_g}{c_h} \right) \mathbb{E}\left[ (H-G)^2 \mid X = x \right]}{\text{Var}(H) - \mathbb{E}[(H-G)^2] + \mathbb{E}\left[ (H-G)^2 \frac{1}{p^\top I} \right] + \frac{\lambda_x}{P_x}}}. \tag{12}$$

Next, we require some detailed case-by-case analysis.

**Case 1: The Interior.** First, we handle the case when the constraint is inactive, i.e., for any fixed $\lambda$, $p_x(\lambda) \in (0, 1)$. (If no such $x$ exists, then the solution is trivially $p(\lambda) = \mathbf{1}_{|\mathcal{X}|}$.) For any $x$ such that $p_x(\lambda)$ is in the interior, by complementary slackness, $\lambda_x = 0$. Now, for any $x' \in \mathcal{X}$ satisfying $p_{x'}(\lambda) \in (0, 1)$, we can write

$$\frac{p_x(\lambda)}{p_{x'}(\lambda)} = \sqrt{\frac{\mathbb{E}\left[ (H-G)^2 \mid X = x \right]}{\mathbb{E}\left[ (H-G)^2 \mid X = x' \right]}},$$

simply by applying (12) to $x$ and $x'$, then dividing these expressions. This tells us that for all $\lambda$ and all $x$ such that $p_x(\lambda)$ is in the interior, $p_x(\lambda) = \gamma u(x)$ for some as-yet-unknown

$$\gamma \in \left( 0, \frac{1}{\sup_{x: p_x \in (0,1)} u(x)} \right].$$

Because $\lambda_x = 0$ on these $x$, the solution to the optimization problem must have the same property.

**Case 2: The Boundary.** When the constraint is active, $p_x(\lambda) = 1$, since $p_x(\lambda) = 0$ is almost always impossible, as established earlier. Examining (12) shows us that the constraint only activates in the case that $u(x) = \mathbb{E}[(H-G)^2 \mid X = x]$ is too large:

$$p_x = 1 \iff u(x) \geq \sqrt{\frac{\text{Var}(H) - \mathbb{E}[(H-G)^2] + \mathbb{E}\left[ (H-G)^2 \frac{1}{p^\top I} \right]}{p^\top P + \frac{c_g}{c_h}}} = \tau(p),$$

since in the alternate case, the unconstrained solution lies in the interior. The Lagrange multiplier $\lambda_x$, in this case, takes on the value such that $p_x(\lambda_x) = 1$; a non-negative such value always exists by virtue of the fact that $u(x)$ is sufficiently large.

Combining Case 1 and Case 2 tells us that our optimal policy has the form

$$\pi(x) = \begin{cases} \gamma \sqrt{u(x)} & \sqrt{u(x)} \leq \tau \\ 1 & \text{otherwise} \end{cases},$$

for a $\tau \in \mathbb{R}_{>0}$ and $\gamma \in \left( 0, \inf_{x: u(x) \leq \tau^2} \sqrt{u(x)} \right]$, which we assume w.l.o.g. is equivalent to $\gamma \in \left( 0, \frac{1}{\tau} \right)$. The constraint on $\gamma$ is necessary, as otherwise we can have $\pi(x) > 1$, which is a contradiction.

Note that another way to express this policy is as $p_x = \mathbb{1}\left\{u(x) > \tau^2\right\} + \gamma\sqrt{u(x)}\mathbb{1}\left\{u(x) \leq \tau^2\right\}$. With this in mind, and defining the vector $U$ with entries $U_x = \mathbb{E}[(H - G)^2 \mid X = x]$ and $W = \text{Var}(H) - \mathbb{E}[(H - G)^2]$ we can rewrite the objective in (11) as

$$J(p) = \left(\sum_{x \in \mathcal{X}} p_x P_x\right)\left(W + \mathbb{E}\left[(H - G)^2\frac{1}{p_X}\right]\right) + \frac{c_g}{c_h}\mathbb{E}\left[(H - G)^2\frac{1}{p_X}\right]$$

which is equivalent to

$$J(\gamma, \tau) = \mathbb{E}\left[\mathbb{1}\left\{U_X > \tau^2\right\} + \gamma\sqrt{U_X}\mathbb{1}\left\{U_X \leq \tau^2\right\}\right]$$
$$\times \left(W + \mathbb{E}\left[(H - G)^2\mathbb{1}\left\{U_X > \tau^2\right\}\right] + \mathbb{E}\left[\frac{(H - G)^2}{\gamma\sqrt{U_X}}\mathbb{1}\left\{U_X \leq \tau^2\right\}\right]\right)$$
$$+ \frac{c_g}{c_h}\left(\mathbb{E}\left[(H - G)^2\mathbb{1}\left\{U_X > \tau^2\right\}\right] + \mathbb{E}\left[\frac{(H - G)^2}{\gamma\sqrt{U_X}}\mathbb{1}\left\{U_X \leq \tau^2\right\}\right]\right)$$

This objective is convex in $\gamma$, but not differentiable or convex in $\tau$. For that reason, we will solve for the optimal $\gamma$ as a function of $\tau$ subject to the constraint that $\gamma > 0$ and $\gamma\sqrt{u(x)} \leq 1 \,\forall x$ where $u(x) \leq \tau^2$, and our algorithm will search over $\tau$ to complete the optimization. Keeping only terms with a dependence on $\gamma$, and recognizing that $\mathbb{E}\left[\frac{(H-G)^2}{\sqrt{U_X}}\right] = \mathbb{E}\left[\sqrt{U_X}\right]$ gives the expression

$$\mathbb{E}\left[\sqrt{U_X}\mathbb{1}\left\{U_X \leq \tau^2\right\}\right] \times$$
$$\left[\frac{1}{\gamma}\left(\mathbb{E}\left[\mathbb{1}\left\{U_X > \tau^2\right\}\right] + \frac{c_g}{c_h}\right) + \gamma\left(W + \mathbb{E}\left[(H - G)^2\mathbb{1}\left\{U_X > \tau^2\right\}\right]\right)\right] \quad (13)$$

Once again, we know that the optimal solution as a function of $\tau$, $\gamma^*(\tau)$ lies either on the boundary or the interior, and we will compare the values of the objective in both cases. In the case that $\gamma^*(\tau) \in (0, \tau^{-1})$, $\gamma^*(\tau)$ is a critical point, thus differentiating and setting equal to zero gives that

$$\frac{1}{\gamma^*(\tau)^2}\left(\frac{c_g}{c_h} + \mathbb{P}\left(U_X > \tau^2\right)\right) = W + \mathbb{E}\left[(H - G)^2\mathbb{1}\left\{U_X > \tau^2\right\}\right],$$

and thus,

$$\gamma^*(\tau)^2 = \frac{\frac{c_g}{c_h} + \mathbb{P}\left(U_X > \tau^2\right)}{W + \mathbb{E}\left[(H - G)^2\mathbb{1}\left\{U_X > \tau^2\right\}\right]} = \frac{\frac{c_g}{c_h} + \mathbb{P}\left(U_X > \tau^2\right)}{\text{Var}(H) - \mathbb{E}\left[(H - G)^2\mathbb{1}\left\{U_X \leq \tau^2\right\}\right]}. \quad (14)$$

As in the proof of Proposition 1, because we are in the case $\gamma^* \in (0, \tau^{-1})$, the right-hand side must be positive and it cannot be greater than $\tau^{-1}$. This gives us that

$$\gamma^*(\tau) \in (0, \tau^{-1}) \implies \mathbb{E}\left[(H - G)^2\mathbb{1}\left\{U_X \leq \tau^2\right\}\right] < \text{Var}(H)$$

and

$$\frac{\frac{c_g}{c_h} + \mathbb{P}\left(U_X > \tau^2\right)}{\text{Var}(H) - \mathbb{E}\left[(H - G)^2\mathbb{1}\left\{U_X \leq \tau^2\right\}\right]} < \frac{1}{\tau^2},$$

and under these conditions we can take square roots on both sides of (14) to obtain

$$\gamma^*(\tau) = \sqrt{\frac{\frac{c_g}{c_h} + \mathbb{P}\left(U_X > \tau^2\right)}{\text{Var}(H) - \mathbb{E}\left[(H - G)^2\mathbb{1}\left\{U_X \leq \tau^2\right\}\right]}} = \sqrt{\frac{\frac{c_g}{c_h} + \mathbb{P}\left(U_X > \tau^2\right)}{\text{Var}(H) - \mathbb{E}\left[U_X\mathbb{1}\left\{U_X \leq \tau^2\right\}\right]}}. \quad (15)$$

Comparing the objective value with $\gamma^*(\tau) = \tau^{-1}$ vs (15), we know that (13) is decreasing in $\gamma$ for $0 < \gamma < \sqrt{\frac{\frac{c_g}{c_h} + \mathbb{P}(U_X > \tau^2)}{\text{Var}(H) - \mathbb{E}[U_X\mathbb{1}\{U_X \leq \tau^2\}]}}$. Thus, we have that

$$\gamma^*(\tau) = \min\left(\sqrt{\frac{c_g/c_h + \mathbb{P}\left(U_X > \tau^2\right)}{\left(\text{Var}(H) - \mathbb{E}[U_X\mathbb{1}\left\{U_X \leq \tau^2\right\}]\right)_+}}, \frac{1}{\tau}\right).$$

Plugging into the original objective $J(\tau, \gamma^*(\tau))$ and minimizing over $\tau$ yields the solution. $\qquad\square$

### C.3 PROOF OF PROPOSITION 4

*Proof.* Since $\mathrm{Error}_T(\pi)$ is monotone in $T$ for all $\pi$, we should first set $T^{\mathrm{stop}}$ to be the largest $T$ for which the constraint holds. This value is

$$T^{\mathrm{stop}} = \left\lfloor \frac{B}{c_h p + c_g} \right\rfloor.$$

Plugging this into the objective yields

$$\frac{1}{\left\lfloor \frac{B}{c_h p + c_g} \right\rfloor} \left( \mathrm{Var}(H) - \mathbb{E}[(H - G)^2] + \frac{1}{p} \mathbb{E}\left[(H - G)^2\right] \right).$$

This is a complicated optimization problem because of the floor function, and cannot be solved by setting the gradient to zero. We will begin by searching over all values of $p \in (0, 1]$ for which $\frac{B}{c_h p + c_g} = k$ for $k \in \mathbb{N}_+$, i.e., $p \in \left\{ \frac{B - k c_g}{k c_h} : k \in \{\lceil B/(c_h + c_g)\rceil, \ldots, \lfloor B/c_g \rfloor\} \right\}$. In terms of $k$, and denoting $E = \mathbb{E}[(H - G)^2]$ and $V = \mathrm{Var}(H) - \mathbb{E}[(H - G)^2]$, the objective then becomes

$$\frac{1}{k} \left( V + \frac{k c_h}{B - k c_g} E \right) = \frac{1}{k} V + \frac{c_h}{B - k c_g} E.$$

Ignoring the discreteness of $k$, in the case that $p^* \in (0, 1)$ we can set the derivative to zero, getting

$$\frac{c_g c_h}{(B - k c_g)^2} E = \frac{1}{k^2} V$$

$$\Longleftrightarrow k^2 c_g c_h \frac{E}{V} = (B - k c_g)^2$$

$$\Longleftrightarrow k^2 \left( c_g^2 - c_g c_h \frac{E}{V} \right) - 2 k c_g B + B^2 = 0$$

The positive solution to this quadratic is

$$k = \frac{2 c_g B + \sqrt{4 c_g^2 B^2 - 4 B^2 \left( c_g^2 - c_g c_h \frac{E}{V} \right)}}{2 \left( c_g^2 - c_g c_h \frac{E}{V} \right)} = B \frac{1 + \sqrt{\frac{c_h}{c_g} \frac{E}{V}}}{c_g - c_h \frac{E}{V}}.$$

Thus, the optimal $k^*$ solves the following optimization problem:

$$k^* = \underset{k \in \left\{ \left\lfloor B \frac{1 + \sqrt{\frac{c_h}{c_g} \frac{E}{V}}}{c_g - c_h \frac{E}{V}} \right\rfloor, \left\lceil B \frac{1 + \sqrt{\frac{c_h}{c_g} \frac{E}{V}}}{c_g - c_h \frac{E}{V}} \right\rceil \right\}}{\mathrm{argmin}} \ \frac{1}{k} V + \frac{c_h}{B - k c_g} E,$$

And the optimal $p^*$ is either

$$p^* = \frac{B - k^* c_g}{k^* c_h}$$

or the boundary solution $p^* = 1$. To disambiguate between the two, we can directly compute the objective value for each. $\square$

### C.4 PROOF OF PROPOSITION 5

*Proof.* The asymptotic normality statement can be read off as a simplified version of Theorem 1 from (Zrnic & Candès, 2024). The second part follows because if $Z \sim \mathcal{N}(0, \Sigma^*)$, then $\|Z\|_2^2 = \|(V^*)^{-1/2} Z\|_2^2$, where $V^*$ is the eigenvector matrix of $\Sigma^*$ (since $(V^*)^{-1/2}$ is unitary). Thus, taking $\Lambda^*$ to be the (diagonal) eigenvalue matrix of $\Sigma^*$ and defining we have that $\|Z\|_2^2 \overset{\mathrm{d}}{=} \|\Lambda Z'\|_2^2$, where $Z' \sim \mathbb{N}(0, \mathbf{I}_d)$. Since $\|\Lambda Z'\|_2^2 = \sum_{j=1}^{d} \lambda_j (Z'_j)^2$, and $Z'_j \overset{\mathrm{i.i.d.}}{\sim} \chi_1^2$, the result is proven. $\square$

### C.5 PROOF OF PROPOSITION 6

*Proof.* Following the simplification of Problem (5), our problem is equivalent to minimizing the following objective:

$$(c_h \mathbb{E}[\pi(X)] + c_g) \operatorname{Tr}(\Sigma^*). \tag{16}$$

Expanding out $\Sigma^*$, we can write

$$\operatorname{Tr}(\Sigma^*) = \operatorname{Tr}\left(W_{\theta^*}^{-1} \operatorname{Var}\left(\nabla \ell_{\theta^*}^g + (\nabla \ell_{\theta^*} - \nabla \ell_{\theta^*}^g)\frac{\xi}{\pi(X)}\right) W_{\theta^*}^{-1}\right)$$

Expanding out the variance gives

$$\operatorname{Var}\left(\nabla \ell_{\theta^*}^g + (\nabla \ell_{\theta^*} - \nabla \ell_{\theta^*}^g)\frac{\xi}{\pi(X)}\right)$$

$$= \mathbb{E}\left[\left(\nabla \ell_{\theta^*}^g + (\nabla \ell_{\theta^*} - \nabla \ell_{\theta^*}^g)\frac{\xi}{\pi(X)}\right)\left(\nabla \ell_{\theta^*}^g + (\nabla \ell_{\theta^*} - \nabla \ell_{\theta^*}^g)\frac{\xi}{\pi(X)}\right)^\top\right] - \mathbb{E}[\nabla \ell_{\theta^*}]\mathbb{E}[\nabla \ell_{\theta^*}]^\top.$$

Expanding out the squared term yields

$$\mathbb{E}\left[\left(\nabla \ell_{\theta^*}^g + (\nabla \ell_{\theta^*} - \nabla \ell_{\theta^*}^g)\frac{\xi}{\pi(X)}\right)\left(\nabla \ell_{\theta^*}^g + (\nabla \ell_{\theta^*} - \nabla \ell_{\theta^*}^g)\frac{\xi}{\pi(X)}\right)^\top\right]$$

$$= \mathbb{E}\left[\nabla \ell_{\theta^*}^g (\nabla \ell_{\theta^*}^g)^\top\right]$$

$$+ \mathbb{E}\left[\frac{\xi}{\pi(X)}\left(\nabla \ell_{\theta^*}^g (\nabla \ell_{\theta^*} - \nabla \ell_{\theta^*}^g)^\top + (\nabla \ell_{\theta^*} - \nabla \ell_{\theta^*}^g)(\nabla \ell_{\theta^*}^g)^\top\right)\right]$$

$$+ \mathbb{E}\left[\left((\nabla \ell_{\theta^*} - \nabla \ell_{\theta^*}^g)\frac{\xi}{\pi(X)}\right)\left((\nabla \ell_{\theta^*} - \nabla \ell_{\theta^*}^g)\frac{\xi}{\pi(X)}\right)^\top\right]$$

$$= \mathbb{E}\left[\nabla \ell_{\theta^*}^g (\nabla \ell_{\theta^*}^g)^\top\right]$$

$$+ \mathbb{E}\left[\nabla \ell_{\theta^*}^g (\nabla \ell_{\theta^*} - \nabla \ell_{\theta^*}^g)^\top + (\nabla \ell_{\theta^*} - \nabla \ell_{\theta^*}^g)(\nabla \ell_{\theta^*}^g)^\top\right]$$

$$+ \mathbb{E}\left[\frac{1}{\pi(X)}((\nabla \ell_{\theta^*} - \nabla \ell_{\theta^*}^g))((\nabla \ell_{\theta^*} - \nabla \ell_{\theta^*}^g))^\top\right].$$

Thus, by the linearity of the Tr operator, we can rewrite the trace as $\operatorname{Tr}(\Sigma^*) = \mathbb{E}\left[\frac{M}{\pi(X)}\right] + C$, where

$$M = \operatorname{Tr}\left(W_{\theta^*}^{-1}(\nabla \ell_{\theta^*} - \nabla \ell_{\theta^*}^g)(\nabla \ell_{\theta^*} - \nabla \ell_{\theta^*}^g)^\top W_{\theta^*}^{-1}\right)$$

and $C$ is

$$\operatorname{Tr}\left(W_{\theta^*}^{-1}\left(\mathbb{E}\left[\nabla \ell_{\theta^*}^g (\nabla \ell_{\theta^*}^g)^\top + \nabla \ell_{\theta^*}^g (\nabla \ell_{\theta^*} - \nabla \ell_{\theta^*}^g)^\top + (\nabla \ell_{\theta^*} - \nabla \ell_{\theta^*}^g)(\nabla \ell_{\theta^*}^g)^\top\right] - \mathbb{E}[\nabla \ell_{\theta^*}]\mathbb{E}[\nabla \ell_{\theta^*}]^\top\right)W_{\theta^*}^{-1}\right)$$

$$= \operatorname{Tr}\left(W_{\theta^*}^{-1}\left(\mathbb{E}\left[\nabla \ell_{\theta^*}^g (\nabla \ell_{\theta^*})^\top + (\nabla \ell_{\theta^*} - \nabla \ell_{\theta^*}^g)(\nabla \ell_{\theta^*}^g)^\top\right] - \mathbb{E}[\nabla \ell_{\theta^*}]\mathbb{E}[\nabla \ell_{\theta^*}]^\top\right)W_{\theta^*}^{-1}\right).$$

Returning to the objective, and excluding factors that do not depend on $\pi$, we can write it now as

$$(c_h \mathbb{E}[\pi(X)] + c_g)\left(\mathbb{E}\left[\frac{M}{\pi(X)}\right] + C\right) \propto_\pi (c_h \mathbb{E}[\pi(X)] + c_g)\mathbb{E}\left[\frac{M}{\pi(X)}\right] + c_h \mathbb{E}[\pi(X)]C.$$

In discrete form, following Propostion 2, this is equivalent to

$$(c_h p^\top P + c_g)\mathbb{E}\left[\frac{M}{p^\top I}\right] + c_h p^\top PC.$$

Taking the derivative with respect to $p$ and setting it to zero coordinatewise yields

$$c_h P_x \mathbb{E}\left[\frac{M}{p^\top I}\right] + c_h P_x C = (c_h p^\top P + c_g)\mathbb{E}[MI_x],$$

and thus,

$$p_x = \sqrt{\frac{(c_h p^\top P + c_g)\mathbb{E}\left[M \mid X = x\right]}{c_h \mathbb{E}\left[\frac{M}{p^\top I}\right] + c_h C}} \propto_x \sqrt{\mathbb{E}\left[M \mid X = x\right]} = \sqrt{U(x)}.$$

Plugging $\pi(x) = \gamma\sqrt{\mathbb{E}\left[M \mid X = x\right]}$ back into (16) gives the one-dimensional objective

$$\frac{c_g}{\gamma}\mathbb{E}\left[\frac{M}{\sqrt{\mathbb{E}\left[M \mid X = x\right]}}\right] + c_h\gamma\mathbb{E}[\sqrt{\mathbb{E}\left[M \mid X = x\right]}]C.$$

The tower property gives us that $\mathbb{E}\left[\frac{M}{\sqrt{\mathbb{E}[M|X=x]}}\right] = \mathbb{E}\left[\sqrt{\mathbb{E}\left[M \mid X = x\right]}\right]$, yielding the objective

$$\frac{c_g}{\gamma}\mathbb{E}\left[\sqrt{\mathbb{E}\left[M \mid X = x\right]}\right] + c_h\gamma\mathbb{E}[\sqrt{\mathbb{E}\left[M \mid X = x\right]}]C,$$

which is equivalent to minimizing

$$\frac{c_g}{\gamma} + c_h\gamma C.$$

The solution to this problem is

$$\gamma^* = \sqrt{\frac{c_g}{c_h} \cdot \frac{1}{C}}.$$

$\square$

### C.6 PROOF OF PROPOSITION 8

*Proof.* Following the derivation in Section B.2, we have that for any $\pi$

$$\mathrm{Var}(\Delta^\pi) = \mathrm{Var}(H) - \mathbb{E}[(H - G)^2] + \mathbb{E}\left[(H - G)^2\frac{1}{\pi(X)}\right].$$

We then immediately get that

$$\mathrm{Var}(\Delta^{\tilde{\pi}}) - \mathrm{Var}(\Delta^{\pi^*}) = \mathbb{E}\left[\frac{(H - G)^2}{\tilde{\pi}(X)} - \frac{(H - G)^2}{\pi^*(X)}\right] \le b\mathbb{E}\left[\frac{1}{\tilde{\pi}(X)} - \frac{1}{\pi^*(X)}\right] \le b\delta.$$

$\square$

### C.7 PROOF OF COROLLARY 9

*Proof.* Since

$$\tilde{\pi}(x) = (\gamma^* + \delta_\gamma)\sqrt{U(x) + \delta_U(x)},$$

we have

$$\frac{1}{\tilde{\pi}(x)} = \frac{1}{(\gamma^* + \delta_\gamma)\sqrt{U(x) + \delta_U(x)}} = \frac{1}{\gamma^*\sqrt{U(x)}}\frac{1}{\left(1 + \frac{\delta_\gamma}{\gamma^*}\right)\sqrt{1 + \frac{\delta_U(x)}{U(x)}}}.$$

A first-order Taylor expansion yields

$$\frac{1}{\left(1 + \frac{\delta_\gamma}{\gamma^*}\right)\sqrt{1 + \frac{\delta_U(x)}{U(x)}}} = 1 - \frac{\delta_\gamma}{\gamma^*} - \frac{1}{2}\frac{\delta_U(x)}{U(x)} + o\left(\delta_\gamma, \frac{\delta_U(x)}{U(x)}\right).$$

Thus,

$$\frac{1}{\tilde{\pi}(x)} - \frac{1}{\pi^*(x)} = \frac{-\delta_\gamma}{(\gamma^*)^2\sqrt{U(x)}} - \frac{1}{2\gamma^*}\frac{\delta_U(x)}{U(x)^{3/2}} + o\left(\delta_\gamma, \frac{\delta_U(x)}{U(x)}\right).$$

Ignoring second-order terms, since $U(x) \ge \epsilon$ almost surely, we have

$$\left|\frac{1}{\tilde{\pi}(x)} - \frac{1}{\pi^*(x)}\right| \le \frac{|\delta_\gamma|}{(\gamma^*)^2\sqrt{\epsilon}} + \frac{1}{2\gamma^*\epsilon^{3/2}}|\delta_U(x)|.$$

Taking the expectation and using linearity,

$$\mathbb{E}\left[\frac{1}{\tilde{\pi}(X)} - \frac{1}{\pi^*(X)}\right] \leq \frac{|\delta_\gamma|}{(\gamma^*)^2\sqrt{\epsilon}} + \frac{1}{2\gamma^*\epsilon^{3/2}}\,\mathbb{E}[\,|\delta_U(X)|].$$

Plugging this bound into the initial inequality for $\mathrm{Var}(\Delta^{\tilde{\pi}})$ completes the proof:

$$\mathrm{Var}(\Delta^{\tilde{\pi}}) - \mathrm{Var}(\Delta^{\pi^*}) \leq b\left(\frac{|\delta_\gamma|}{(\gamma^*)^2\sqrt{\epsilon}} + \frac{1}{2\gamma^*\epsilon^{3/2}}\,\mathbb{E}[\,|\delta_U(X)|]\right).$$

$\square$

## C.8 PROOF OF PROPOSITION 10

*Proof.* We will borrow notation from the proof of Proposition 2, and express all quantities in vector form. The optimization problem in (9) only depends on $Q$ through the likelihood ratio, $\frac{dP}{dQ} = r \in \mathbb{R}^{|\mathcal{X}|}_{>0}$, and $Q, P$ are absolutely continuous with respect to one another. So, we will learn $r$ and then calculate $Q^* = P/r$.

Ignoring terms that do not depend on $r$, the problem in (9) can be rewritten as

$$\underset{r\in\mathbb{R}^{|\mathcal{X}|}_{>0}}{\text{minimize}} \quad r^\top \mathbb{E}_P\left[I\left(H^2 + \left(\frac{1}{\pi(X)} - 1\right)(H - G)^2\right)\right]$$

$$\text{subject to} \quad (1/r)^\top P = 1.$$

Forming the Lagrangian,

$$\mathcal{L}(r, \lambda) = r^\top \mathbb{E}_P\left[I\left(H^2 + \left(\frac{1}{\pi(X)} - 1\right)(H - G)^2\right)\right] + \lambda((1/r)^\top P - 1).$$

Taking the gradient gives

$$\nabla_r \mathcal{L}(r, \lambda) = \mathbb{E}_P\left[I\left(H^2 + \left(\frac{1}{\pi(X)} - 1\right)(H - G)^2\right)\right] - \lambda P/(r^2),$$

and setting it to zero yields

$$r_x^* \propto_x \sqrt{\frac{1}{\mathbb{E}_P\left[\left(H^2 + \left(\frac{1}{\pi(X)} - 1\right)(H - G)^2\right)\Big|X = x\right]}} = \sqrt{\frac{1}{\nu_x}}.$$

To ensure the proper normalization, we set

$$r_x^* = \frac{\sqrt{\nu}^\top P}{\sqrt{\nu_x}}.$$

Thus, $Q^*(x) = P/r^* = \frac{\sqrt{\nu_x}P_x}{\sqrt{\nu}^\top P}$.

$\square$

## D ADDITIONAL EMPIRICAL RESULTS

### D.1 BERNOULLI DATA: COMPARING $\pi_{\text{active}}$ TO $\pi_{\text{random}}$

Figure 4 provides additional results for the Bernoulli data setting in Section 3.3 when comparing $\pi_{\text{active}}$ to $\pi_{\text{random}}$. Recall that here the results differ from comparing to $\pi_{\text{base}}$ only when $\mathrm{MSE}(H, G) < \frac{c_h}{c_h + c_g}\mathrm{Var}(H)$, as otherwise the optimal sampling rate for $\pi_{\text{random}}$ is simply $p^* = 1$.

#### D.1.1 ON THE ERROR RATIO LOWER BOUND

It is interesting to observe that $\mathrm{ErrorRatio}(\pi_{\text{active}}, \pi_{\text{base}})$ is lower-bounded in the Bernoulli data setting at a value close to $\mathrm{MSE}(H, G)$. To see why, we note that the lowest value of $\mathrm{ErrorRatio}(\pi_{\text{active}}, \pi_{\text{base}})$ is obtained when $U$ is maximum variance—which is achieved when $U$

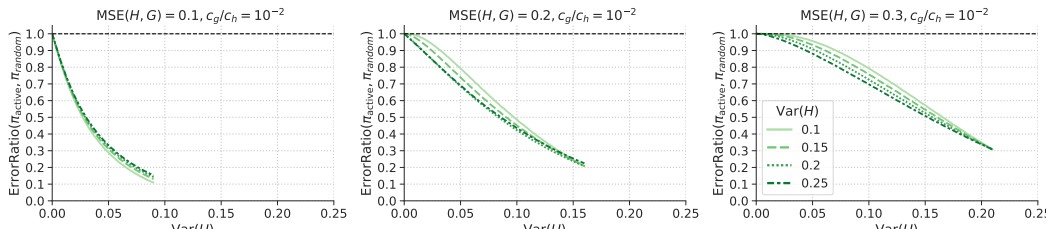

Figure 4: Results on the Bernoulli data (§3.3) for $\pi_{\text{active}}$ vs. $\pi_{\text{random}}$ while varying $\text{MSE}(H, G)$ and $\text{Var}(U)$. As in Figure 1, each line plots a different value of $\text{Var}(H)$, where we choose values that are representative of low, medium, or high variance settings compared to $\text{MSE}(H, G)$.

is a binary random variable that is 1 when $G \neq H$, and 0 otherwise. Recall that in the Bernoulli data setting both $H$ and $G$ are binary, and $\text{MSE}(H, G) = \mathbb{P}(H \neq G)$. We can then compute $\text{ErrorRatio}(\pi_{\text{active}}, \pi_{\text{base}})$ after optimizing over $\tau$ as approaching

$$\min \left( \frac{\left( \gamma \text{MSE}(H, G) + \frac{c_g}{c_h} \right) \left( 1 + (\frac{1}{\gamma} - 1) \frac{\text{MSE}(H,G)}{\text{Var}(H)} \right)}{\text{MSE}(H, G) + \frac{c_g}{c_h}} \right)$$

$$\text{where } \gamma = \sqrt{\frac{c_g/c_h}{(\text{Var}(H) - \text{MSE}(H, G))_+}}.$$

Note that we have the first quantity only when $\text{MSE}(H, G) \leq \text{Var}(H) + c_g/c_h$.

*Derivation.* When $U \to \mathbb{1}\{H \neq G\} \in \{0, 1\}$, from Proposition (2) $\pi_{\text{active}}$ approaches either

$$\pi_{\text{clip}}(x, \tau = 1) = \begin{cases} \gamma^*(1) & \text{if } h(x) \neq g(x) \\ 0 & \text{otherwise} \end{cases} \quad \text{or} \quad \pi_{\text{clip}}(x, \tau = 0) = \begin{cases} 1 & \text{if } h(x) \neq g(x) \\ 0 & \text{otherwise} \end{cases}$$

where $\gamma^*(1) = \sqrt{\frac{c_g/c_h}{(\text{Var}(H) - \text{MSE}(H,G))_+}}$. Plugging these values into the optimization over $\tau \in \{0, 1\}$,

$$\tau^* = \underset{\tau \in \{0,1\}}{\operatorname{argmin}} \left( c_h \mathbb{E}[\pi_{\text{clip}}(x; \tau)] + c_g \right) \left( \text{Var}(H) + \mathbb{E}\left[ U \left( \pi_{\text{clip}}(x; \tau)^{-1} - 1 \right) \right] \right),$$

at $\tau = 1$ we get

$$\left( c_h \gamma^*(1) \text{MSE}(H, G) + c_g \right) \left( \text{Var}(H) + \left( \frac{1}{\gamma^*(1)} - 1 \right) \text{MSE}(H, G) \right),$$

and at $\tau = 0$ we get

$$\left( c_h \text{MSE}(H, G) + c_g \right) \text{Var}(H),$$

so the optimal $\tau^*$ is the smaller of the two. Dividing each by $c_h \text{Var}(H)$ and taking the minimum gives the result for $\text{ErrorRatio}(\pi_{\text{active}}, \pi_{\text{base}})$. $\square$

A similar calculation can also be made for $\text{ErrorRatio}(\pi_{\text{active}}, \pi_{\text{random}})$, with different bounds for when $\text{MSE}(H, G) \leq \text{Var}(H) - \frac{c_g}{c_h}$ and/or $\text{MSE}(H, G) \leq \frac{c_h}{c_g + c_h} \text{Var}(H)$ (i.e., both conditions, one or the other condition, or neither condition).

## D.2 ADDITIONAL DATASETS

We provide experimental results on two additional datasets:

**ImageNet.** The ImageNet dataset (Deng et al., 2009) categorizes input images into one of $1k$ classes. Our goal is to evaluate the accuracy $\mathbb{E}[H]$ of a pretrained ResNet model (He et al., 2016), where $H$ is the binary indicator of whether the model's prediction matches the human label for a given image $X$. $G$ is the softmax value the model assigns to its predicted class. $U$ is computed as $G(1 - G)$.

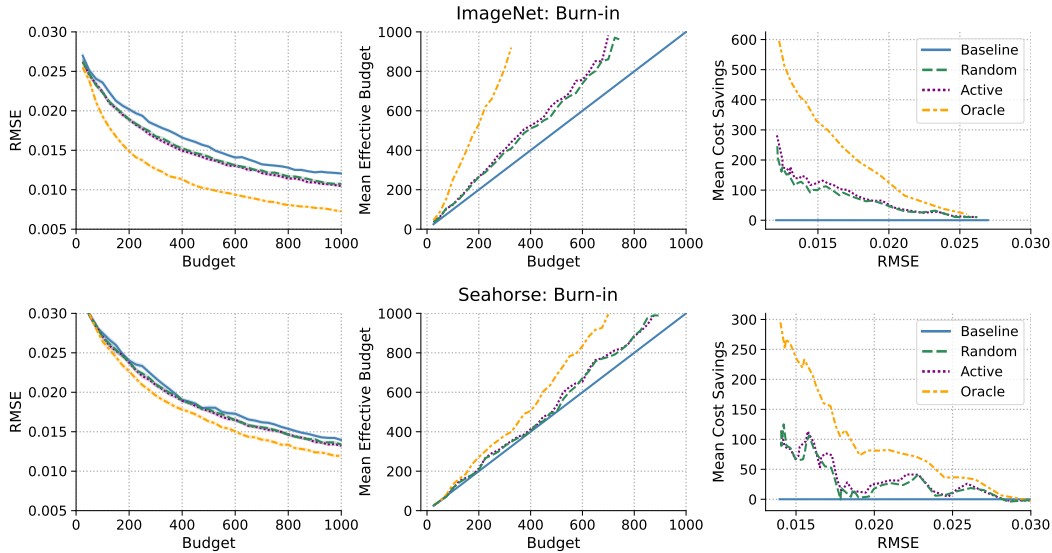

Figure 5: Results on the ImageNet and Seahorse datasets using 200 examples as a burn-in (approach A2 in Section 4). The budget on the x-axis reflects "additional" budget used *after* the burn-in examples.

**Seahorse.** The Seahorse dataset (Clark et al., 2023) focuses on multilingual summarization. We focus on the "attribution to the source document" metric for summaries produced by a finetuned 13B parameter mT5 model (Xue et al., 2020). $H$ comes from human ratings. $G$ is the probability score from a finetuned mT5-XXL autorater model assessing attribution.[3] $U$ is computed as $G(1 - G)$.

Results are shown in Figure 5, with similar takeaways as the other burn-in (approach A2) experiments in Section 4.3. For ImageNet, both $\pi_{\text{active}}$ and $\pi_{\text{random}}$ substantially outperform $\pi_{\text{base}}$; however, the estimated $\pi_{\text{active}}$ still leaves a significant amount of headroom behind with respect to the oracle active policy, and has comparable performance to $\pi_{\text{random}}$. The Seahorse dataset is an interesting case where the weak rater $G$ is simply not that good, even conditionally. This results in small (but still positive) gains for both the active and random policies—even when $\pi_{\text{active}}$ uses oracle parameters. This highlights an outcome also understood from our theory: the weak rater $G$ has to be good enough at least some of the time to achieve substantial lift in evaluation efficiency.

### D.3 RELATIVE ERROR AND COST SAVINGS

Figures 6 and 7 plot additional metrics comparing performance to the classical baseline. Specifically:

1. **Percent reduction in RMSE:** The percent reduction in RMSE at a given budget relative to the baseline's RMSE at the same budget.

2. **Percent reduction in cost:** The percent reduction in the budget required by the policy to achieve a given RMSE relative to budget required by the baseline to acheive the same RMSE.

3. **Absolute cost savings:** The difference in the budget required by the baseline to achieve a given RMSE versus the budget required by the policy to acheive the same RMSE.

---

[3]This checkpoint is available at
https://huggingface.co/collections/google/seahorse-release-6543b0c06d87d83c6d24193b

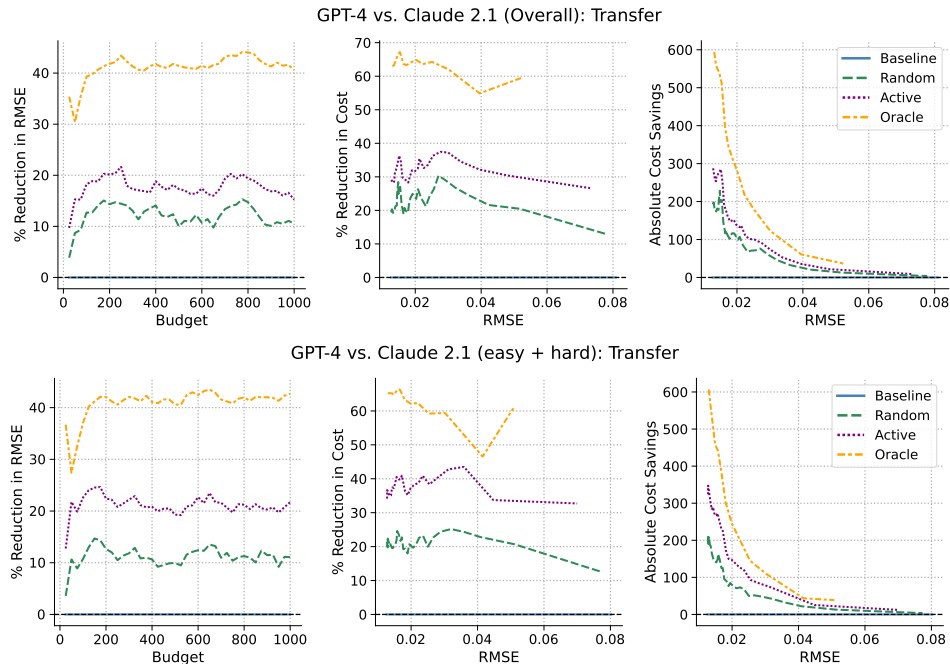

Figure 6: Relative error reduction and cost savings when using policy transfer (Approach A1 in §4).

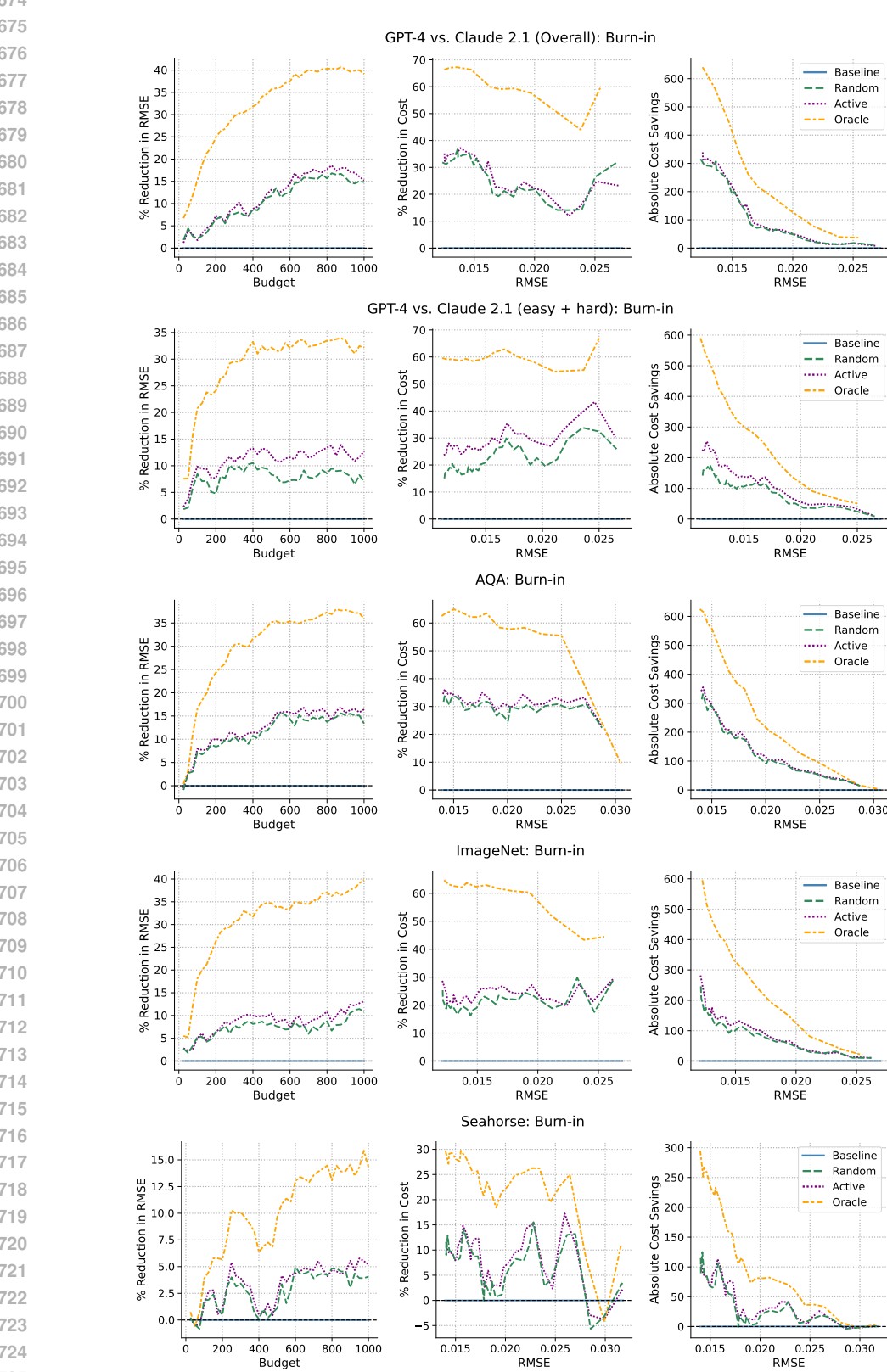

Figure 7: Relative error reduction and cost savings when using burn-in (Approach A2 in §4).

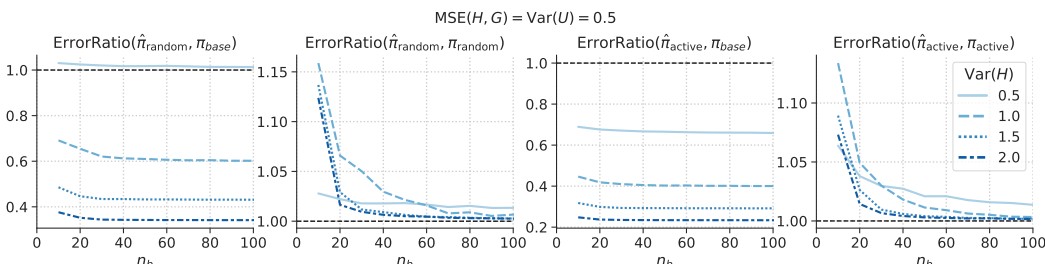

Figure 8: $n_b$ is the number of empirical samples used to estimate policy parameters for $\hat{\pi}_{\text{random}}$ and $\hat{\pi}_{\text{active}}$. $\pi_{\text{random}}$ and $\pi_{\text{active}}$ use the true, optimal parameters. Both $\text{ErrorRatio}(\hat{\pi}_{\text{random}}, \pi_{\text{random}})$ and $\text{ErrorRatio}(\hat{\pi}_{\text{active}}, \pi_{\text{active}})$ converge to $1.0$ with modestly sized $n_b$. See §D.4.1.

### D.4  EFFECT OF THE BURN-IN SIZE $n_b$

The burn-in approach to estimating policy parameters (Approach A2 in Section 4) relies on using the first $n_b$ examples to the strong rater, which at small budgets (i.e., $B \approx n_b$) can be either (a) infeasible, or (b) reduce net efficiency gains. On the other hand, if $n_b$ is too small, policy parameters might be estimated inaccurately. In this section we present an ablation of how the size of the burn-in set affects policy performance: in short, we find that policy performance quickly converges with $n_b$.

#### D.4.1  GAUSSIAN DATA

Using the same Gaussian data setup from Section 3.2, we explore the effects of *estimating* policy parameters using various finite samples of size $n_b$. Specifically, we compute the error ratios of $\hat{\pi}_{\text{active}}$ vs. $\pi_{\text{base}}$ and $\pi_{\text{active}}$, where $\hat{\pi}_{\text{active}}$ uses estimated parameters and $\pi_{\text{active}}$ uses oracle parameters. Note that the finite data is only used to estimate $\gamma^*(\tau)$ and $\tau^*$; we use the uncertainty estimates $u(x)$ as given. We do the same for $\hat{\pi}_{\text{random}}$ vs $\pi_{\text{base}}$ and $\pi_{\text{random}}$, where the $n_b$ data points are used to estimate the fixed sampling rate $p$. Results are shown in Figure 8 for various settings of Var(H) and fixed true MSE(H, G) = Var(U) = 0.5. By $n_b = 20$, the error ratios of both $\hat{\pi}_{\text{random}}$ and $\hat{\pi}_{\text{active}}$ with their respective oracle versions are within $1.10$ for all tested values of $\text{Var}(H)$; by $n_b = 40$ both error ratios are well within $1.05$. By $n_b = 50$, performance relative to the baseline has largely stabilized.

#### D.4.2  REAL DATA

Figure 9 plots results for various burn-in budgets $n_b$ on the Chatbot Arena, AQA, ImageNet, and Seahorse datasets (we plot the same relative metrics as in Appendix D.3). Recall that on these real data collections, the burn-in data is used both the estimate policy parameters, and to calibrate $G$. The burn-in data is also used to warm-start the mean estimate per the policy combination equation in (6). All of these factors are affected by $n_b$. To see these effects more clearly, we plot the *total budget used* on the x-axis, which includes both $n_b$ and any additional budget used thereafter (in increments of 50). The left column then plots the percent reduction in average RMSE achieved at a given total budget over the baseline policy. The middle column plots the percent reduction in average total cost required to reach a certain target RMSE level over the baseline, while the right column plots the absolute savings in average total cost over the baseline. As can be expected, gains over the baseline don't start until the total budget is $> n_b$. However, after that point, the active policy begins to improve considerably over the baseline for all datasets. The datasets with higher $n_b$ also tend to improve at a faster rate over the baseline—likely due to better estimated parameters obtained from the larger burn-in sample size. Smaller $n_b$ sizes also do well over the baseline, though can suffer slightly when $n_b < 100$, and can result in higher variance performance (additionally, we found that $n_b$ below 50 results in unstable, poor performance). Combined, these results suggest that a higher $n_b$ is worth it if the total budget is high and the desired RMSE is low: otherwise there is a tradeoff. Promising directions for future work include deriving an optimal burn-in size, or developing adaptive burn-in strategies.[4]

---

[4]We thank anonymous reviewer MVJi for the suggestion.

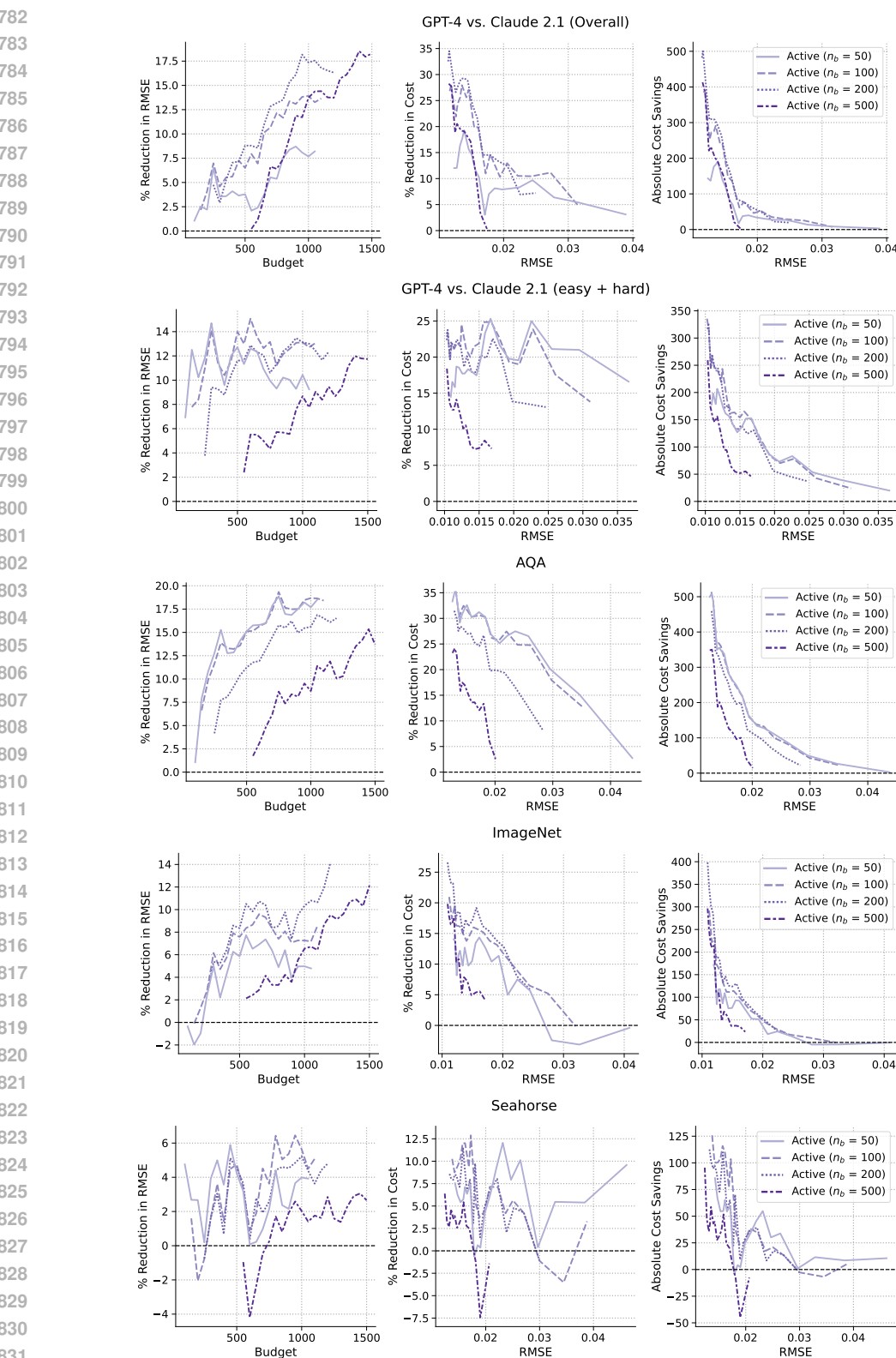

Figure 9: Results when varying the number of samples $n_b$ used for burn-in following approach A2 from §4. Here, the x-axis plots the *total budget used*, which includes the samples used for burn-in.

### D.5 OPTIMIZING $\tau$

The cost-optimal active policy derived in Proposition 2 depends on solving the following 1-d objective for the clipping threshold $\tau^*$:

$$\tau^* = \underset{\tau \in \mathbb{R}_{>0}}{\arg\min} \left( c_h \mathbb{E}[\pi_{\text{clip}}(x; \tau)] + c_g \right) \left( \text{Var}(H) + \mathbb{E}\left[ U \left( \pi_{\text{clip}}(x; \tau)^{-1} - 1 \right) \right] \right).$$

As described in Remark 3, this optimization problem is non-convex and has no analytical solution. However, because it is only 1-d, it is easy to find via a simple grid-search of the objective. In practice, we find $\tau^*$ by defining a coarse grid of quantiles of $U$ (taken from the empirical data used in either approach A1 or A2 from Section 4). To look at the effects of the grid size on performance, Figure **??** plots the error ratio of $\hat{\pi}_{\text{active}}$ vs. $\pi_{\text{active}}$ on the Gaussian data setup from Section 3.2 as a function of the grid size used to search for $\tau$. The oracle comparison, $\pi_{\text{active}}$, uses a grid size of 1000. In this setting, a grid size of 10 is sufficient to match performance of.

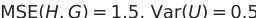

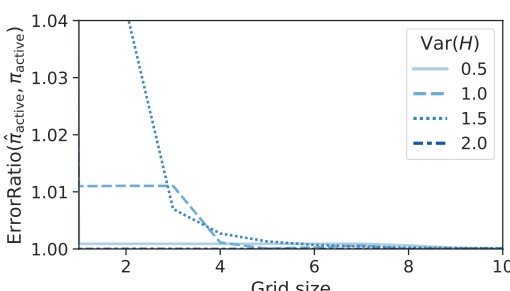

Figure 10: Varying the grid size used to search for the clipping threshold $\tau^*$.

## E IMPLEMENTATION DETAILS

All real data experiments in this paper were performed retrospectively with $G$ and $H$ computed once offline for all inputs $x$ in each dataset. Pretrained models and labels for $G$ and $H$, respectively, were used for all datasets except Chatbot Arena (Section 4).[5] All subsequent experiments for active sampling were then performed on CPU resources with 32GB of RAM.

For the Chatbot Arena dataset, we sampled responses from Gemini 1.5 Flash (Gemini Team, 2024) using an adapted version of the Chatbot Arena auto-eval prompt.[6] Below is an example prompt. Color is added for clarity. Ten responses from Gemini 1.5 Flash are then sampled, with five responses using the same prompt with the order of A and B flipped. The final label is taken as the majority vote.

For $G$, we finetune Gemma-3 4B for two hours on TPUv3 resources. The same prompt is used, however, we maximize the log-likelihood of the target Gemini-based answer used for $H$ instead of sampling. Early stopping is done based on the validation loss at predicting $H$ using a held-out split of the training data (recall that the training data is composed of other model comparisons from the Chatbot Arena dataset that are distinct from the one on which we evaluate our annotation policies).

---

[5]The checkpoint used for Seahorse is available at
https://huggingface.co/collections/google/seahorse-release-6543b0c06d87d83c6d24193b
[6]This prompt is available at
https://github.com/lm-sys/arena-hard-auto/blob/main/config/judge_config.yaml

Please act as an impartial judge and evaluate the quality of the responses provided by two AI Assistants to the user prompt displayed below. You will be given Assistant A's answer and Assistant B's answer. Your job is to evaluate which assistant's answer is better.

When evaluating the assistants' answers, first identify any mistakes or inaccurate information. Next, consider if the assistant's answers are helpful, relevant, and concise. Helpful means the answer correctly responds to the prompt or follows the instructions. Note that when the user prompt has any ambiguity or more than one interpretation, it is more helpful and appropriate to ask for clarifications or more information from the user than providing an answer based on assumptions. Relevant means all parts of the response closely connect or are appropriate to what is being asked. Concise means the response is clear and not verbose or excessive. Then consider the creativity and novelty of the assistant's answers when needed. Finally, identify any missing important information in the assistants' answers that would be beneficial to include when responding to the user prompt.

<|User Prompt|>
make a haiku on bacon the food
<|The End of User Prompt|>

<|The Start of Assistant A's Answer|>
Crisp strips of delight,
Sizzling dance, morning's first light,
Bacon whispers, "Bite."
<|The End of Assistant A's Answer|>

<|The Start of Assistant B's Answer|>
Here is a haiku about bacon:

Sizzling in pan
Savory salty bacon strips
Crispy delight yum
<|The End of Assistant B's Answer|>

Is the higher quality response:
(A) Assistant A is better
(B) Assistant B is better
Please answer with either (A) or (B).

