# OpenReview forum: "Cost-Optimal Active AI Model Evaluation"
_ICLR.cc/2026/Conference — Submitted to ICLR 2026_

### Official Review · Reviewer_te8E · 2025-10-26

**Soundness:** 2
**Presentation:** 2
**Contribution:** 3
**Rating:** 4
**Confidence:** 2

**Summary:**

This paper proposes a framework for the so-called cost-optimal active evaluations, which encompasses a family of annotation policies designed to minimize expected error under a given annotation budget.

**Strengths:**

- The paper presents numerous mathematical propositions to support its claims, demonstrating strong theoretical foundations.
- The motivation behind the work is clearly articulated.
- Key findings are well-highlighted. In particular, the conclusion on line 223—“...active learning can help if the conditional squared error of G has significant variance”—offers valuable guidance for future research in developing active learning strategies.

**Weaknesses:**

Several expressions are vague and fall short of academic standards.

1. For instance, the phrase “...by optimizing everything” in line 086 is unclear—what does “everything” refer to?
The term “evals” used in line 105 and elsewhere is not standard English and lacks a clear definition. The sentence “We now describe our methods for constructing active, cost-optimal evals” (line 105) is difficult to interpret—what exactly are “evals”?
In line 117, H and G are defined as h(X) and G(X), which are described as ratings in line 116. However, line 118 states “querying H and G costs c_h and c_g”, which translates to “querying ratings costs...”—a phrasing that is not easily interpretable.


1. Some mathematical steps are omitted, making the evaluation of the work challenging. For example, the proof of Equation 2 is relegated to the appendix, and the transition from line 726 to line 728 lacks clarity.
2. The term “cost-optimal” used in the title and throughout the paper is not entirely convincing or appropriate. As acknowledged by the authors in line 480—“...annotation policies that are optimal in theory are distribution-dependent...”—this suggests that such optimal policies may be unattainable due to inherent uncertainties. Therefore, the proposed framework does not achieve a truly optimal solution, but rather an optimal solution subject to specific constraints. These constraints should be clearly emphasized, as the current phrasing implies a globally optimal solution.
3. To substantiate the claim in line 477—“We derive annotation policies that are optimal in the sense of minimizing expected error under annotation budget constraints”—a brute-force experiment exploring various combinations of examples and demonstrating that the proposed method achieves the best or ceiling performance would be necessary.

**Questions:**

1. What is the justification for using ξ_t​ over π_t​ in Equation 1? What are the implications of this choice?
2. Should coreset-based methods be included as one of the baselines? If not, why?

---

> ### Author Response · Authors · 2025-11-19
> **Author response**
>
> Thank you for the review! We provide answers to specific questions and remarks (quoted) below.
>
>
> > **Weakness 1:** Several expressions are vague and fall short of academic standards. For instance, the phrase “...by optimizing everything”... The term “evals”...
>
>
> We have updated the language to be more precise, thank you for the suggestion. Please see the revised draft.
>
> > **Weakness 2:** Some mathematical steps are omitted, making the evaluation of the work challenging. For example, the proof of Equation 2 is relegated to the appendix, and the transition from line 726 to line 728 lacks clarity.
>
>
> We have added additional intermediate steps to explicitly show the short calculation required to go from line 726 to 728. Similarly, we added the derivation that shows that the estimator is unbiased for any choice of annotation policy, which we claimed, but did not explicitly show in the main paper. We believe the remainder of the mathematical derivations are clear—and added additional references in the main text that point the reader to where to find them in the Appendix—but please let us know if there are other areas that require explanation.
>
>
> > **Weakness 3:** The term “cost-optimal” used in the title and throughout the paper is not entirely convincing or appropriate...
>
>
> **Propositions 1 and 2 are key results, and they do explicitly specify what the cost-optimal annotation policies are for active model evaluation. In and of themselves, we believe they are substantial, new contributions to the theory of active prediction-powered model evaluation.** That said, yes, our results show that the oracle policies depend on population quantities which must be estimated in practice. These practical approximations of the theoretically optimal policies are the ones that we present in Section 4, and we show that they can achieve the same estimation precision at a far lower total annotation budget than standard evaluation methods, especially in tasks where there is high variability in the difficulty of examples. This is stated in both the abstract and introduction; we have added boldfacing in the revision to make this even clearer.
>
>
> > **Weakness 4:** To substantiate the claim in line 477... a brute-force experiment exploring various combinations of examples and demonstrating that the proposed method achieves the best or ceiling performance would be necessary.
>
>
> We respectfully disagree on this point. The claim that the policies proposed in Section 2 are optimal is rigorously proved.  Our theoretical derivations  (Propositions 1 and 2)  are the necessary and sufficient substantiation for the oracle policies, and the experiments in Section 3 and 4 empirically validate the framework's effectiveness when approximations of these oracles are used instead.
>
>
> > **Question 1:** What is the justification for using $\xi_t$ over $\pi_t$ in Equation 1? What are the implications of this choice?
>
>
> This is a fundamental requirement for the estimator to be unbiased. $\pi_t(X_t)$ is the probability of querying the expensive rater $H_t$ given the input $X_t$. $\xi_t$ is the binary indicator variable (1 or 0) for whether $H_t$ was actually sampled. The structure of the estimator in Equation 1 is the canonical Inverse Probability Weighted (IPW) estimator: dividing by the sampling probability $\pi_t(X_t)$ ensures that $\hat{\theta}_T$ is unbiased for the true mean $\theta^* = \mathbb{E}[H]$. We have added the short calculation that shows this to the Appendix for clarity (see Appendix B.1).
>
>
> > **Question 2:** Should coreset-based methods be included as one of the baselines? If not, why?
>
>
> No, we do not believe that they are relevant baselines. Coreset-based methods typically focus on selecting data points for efficient training and dataset compression, rather than deciding when to use weak labels instead of strong labels. Traditional coreset methods also require labels for the whole dataset; we are not aware of label-free methods appropriate for our problem setting and desiderata. Our framework is focused on unbiased, cost-optimal statistical inference of a target estimand, specifically the estimation of the mean target rating $\mathbb{E}[H]$, by deciding when to use cheap vs. expensive labels.

---

> > ### Comment · Reviewer_te8E · 2025-11-19
> > **Intial Response to Author Response**
> >
> > Thank you for your response. It clarified many of the questions I had.
> >
> > Regarding Weakness 3:
> >
> > Here I do not have any problem understanding the paper or its claims, and I acknowledge that the constraints (e.g., estimating data quantities) may be unavoidable. My concern is specifically about the way the term “cost-optimal” is used in the title and throughout the paper. My initial interpretation was that the method delivers an unconstrained cost-optimal solution, so I was surprised to find that the optimization is actually defined under explicit constraints. I would not have any concern if the phrasing were something along the lines of “cost-aware” or “cost-optimized under certain constraints.”
> >
> > Can you confirm whether my understanding is correct that the model produces an optimal solution given specific constraints? And do you consider this scenario appropriate to describe as a “cost-optimal” solution?
> >
> > Regarding Weakness 4:
> >
> > I have no issue with claiming a theoretically optimal solution. In such a case, experiments may not be necessary if the empirical data perfectly meet the assumptions (e.g., estimate population quantities you noted in the response to Weakness 3). However, there is no guarantee that empirical data will be ideal.
> >
> > Finally, you didn't highlight all edits. I noticed that the revised claim in the conclusion (current line 525 vs. original line 477) was modified without being highlighted. I wouldn't have known have I not double checked with the original manuscript.

---

> > > ### Author Response · Authors · 2025-11-20
> > >
> > > Thank you for your quick reply! We appreciate the opportunity to clarify these remaining points.
> > >
> > > Regarding "cost-optimal," your understanding is correct: we optimize our annotation policies subject to the constraint that the expected cost of the evaluation is below a budget B (this is the informal objective posed in lines 57-59 of the revised draft, and formally stated in Eqs. 3 and 5). We do consider calling the solution to this cost-constrained optimization problem a “cost-optimal policy” to be appropriate, but agree that a more precise title can be useful in avoiding any misalignment with initial expectations. In the updated revision, we have changed the title to “Optimal, Active, Prediction-powered AI Model Evaluation on a Budget” to (a) emphasize the budget constraint, and also (b) reflect that our method leverages PPI to do the evaluation. See also the revised line 53-54. We welcome any further feedback. We have also highlighted this change to the other reviewers in the general response.
> > >
> > > Regarding the gap between theory and empirical data: our experiments are designed to show that even though real-world data may not perfectly match our theoretical assumptions (and in which case we no longer make any claims towards optimality), practical approximations of our policies can still yield substantial cost savings. We’ve made a note that the estimated policies are no longer guaranteed to be optimal (but still perform well). In fact, the performance gap between the "Oracle" and "Active" policies, which we discuss in the paper, is intended to highlight this particular point: that while our practical methods are not perfect, they are a significant step towards more efficient evaluation, and that the remaining gap is a clear area for future work.
> > >
> > > Finally, we apologize for the unhighlighted edits; we tried to highlight the most important changes, but missed some minor revisions.  Thank you for your careful reading. We’ve uploaded a new draft.
> > >
> > > We hope these responses address your remaining concerns—and again, thank you for your engagement.

---

> > > > ### Comment · Reviewer_te8E · 2025-11-20
> > > > **Second Response to Author's following Response**
> > > >
> > > > Thank you for the clarification. I don't have further questions at this moment.

---

### Official Review · Reviewer_BNqA · 2025-10-28

**Soundness:** 3
**Presentation:** 3
**Contribution:** 2
**Rating:** 6
**Confidence:** 2

**Summary:**

This paper introduces a theoretical and empirical framework for cost-optimal active evaluation of generative AI systems. The authors tackle a very practical problem: evaluating large models is expensive, and existing hybrid setups (like combining cheap model raters with expensive human raters) often lack rigorous cost-aware allocation.
To address this, the paper derives policies for optimally allocating annotation budget between weak and strong raters — balancing cost and accuracy through statistical optimization. It extends prediction-powered inference (PPI) and active statistical inference to derive (1) an optimal random sampling rate and (2) an optimal active policy that depends on task-specific uncertainty.

**Strengths:**

1. The problem—cost-aware AI evaluation—is timely, practical, and underexplored. The authors correctly identify inefficiencies in current model evaluation practices that rely heavily on costly human or LLM raters.
2. The extension of prediction-powered inference with explicit cost constraints is technically sound. The derivation of closed-form policies (Propositions 1–2) is clear and builds on well-established statistical theory.
3. The Gaussian/Bernoulli experiments in Section 3 are carefully designed to test key intuitions (e.g., dependence on rater error, heteroskedasticity, and cost ratio). The figures are clean and reinforce the theoretical claims.
4. Applying the framework to Chatbot Arena evaluations shows that cost-optimized sampling can indeed save budget while maintaining accuracy. The setup is realistic and relevant to modern LLM benchmarking.

**Weaknesses:**

1. The real-world experiments are narrow. Most results are on one dataset (Chatbot Arena) with two scenarios, both focused on text-based preference evaluations. There’s little diversity in task type or domain (e.g., no multimodal or structured data). The empirical results, while consistent, are modest—often showing ~40–50% budget savings under ideal transfer, which may shrink with realistic uncertainty estimation.
2. While theoretically elegant, the framework’s impact on real-world evaluation pipelines is unclear. Implementing cost-optimal policies requires calibration, pilot estimation, and maintenance that may offset cost savings in small-to-medium-scale evaluation scenarios.

**Questions:**

1. How do these methods perform when the weak rater is itself biased rather than merely noisy?
2. Are there concrete examples of how much “burn-in” cost is acceptable before cost savings emerge?

---

> ### Author Response · Authors · 2025-11-19
> **Author response**
>
> Thank you for the review! We provide answers to specific questions and remarks (quoted) below.
>
>
> > **Weakness 1:** The real-world experiments are narrow. Most results are on one dataset (Chatbot Arena)... The empirical results, while consistent, are modest—often showing $\approx 40–50%$ budget savings under ideal transfer...
>
>
> In our submission, Appendix D contained additional experimental results on three other datasets (including on ImageNet, another modality beyond text). With the extra space, we have moved one of those datasets (Attributed Question Answering) up to the main paper. The savings are also substantial, especially when considering performance at higher desired precision (i.e., lower MSE estimates). To help understand the relative gains, we have added plots to Appendix D (see Figure 6) that show the percent reduction in RMSE over the baseline at a given budget, as well as the percent reduction in cost required to achieve a target RMSE for **the policies with estimated parameters.** While not as strong as the oracle (as can be expected), the estimated active policy can still achieve **over 35% and over 40% reductions in cost** on the Chatbot Arena dataset in the overall and heterogeneous transfer settings, respectively.
>
>
> > **Weakness 2:** While theoretically elegant, the framework’s impact on real-world evaluation pipelines is unclear. Implementing cost-optimal policies requires calibration, pilot estimation, and maintenance that may offset cost savings...
>
>
> It is true that cost-optimal policies have more moving parts than classical evaluation methods. That said, **the cost benefits can be substantial,** as we show in our experiments. Note that these cost benefits compound as evaluations are repeated again and again over time at large scales, which is true of the current model development lifecycle). Furthermore, the cost-optimal fixed sampling rate policy $\pi_\mathrm{random}$ is also quite simple to implement, and yields consistent savings. For these reasons, we believe that these results can be quite useful for the community.
>
>
> > **Question 1:** How do these methods perform when the weak rater is itself biased rather than merely noisy?
>
>
> Our estimator gives unbiased estimates of $\theta^*$ for any choice of $G$, not just unbiased $G$. However, the performance of the estimator depends on how good of a predictor of $H$ the weak rater $G$ is (this is intuitive, but formally shown in our paper). Specifically, a key quantity is how $E[(H - G)^2]$ compares to $Var(H)$, and we can decompose $E[(H - G)^2]$ as $(E[H] - E[G])^2 + Var(G) + Var(H) - 2Cov(H, G)$. Thus, having a biased weak rater is OK in itself, but if the bias is large and the correlation between H and G is low, then $E[(H - G)^2]$ might be too large for active annotation to be helpful.
>
>
> > **Question 2:** Are there concrete examples of how much “burn-in” cost is acceptable before cost savings emerge?
>
>
> This is a great question, and we have added an entirely new section in Appendix D.4 to address it. In general, the policies do not require $n_b$ to be that large (experiments on the Gaussian data setting in D.4.1 show that the estimated policies converge quite quickly). To directly answer your question about when cost savings begin to emerge, experiments on the real datasets in Section D.4.2 show that gains over the baseline (obviously) don't start until the total budget is $>n_b$. However, after that point, the active policy begins to improve considerably over the baseline for all datasets. The datasets with higher $n_b$ also tend to improve at a faster rate over the baseline---likely due to better estimated parameters obtained from the larger burn-in sample. Using smaller $n_b$ still results in significantly tighter CIs than the baseline, though can suffer slightly from high variance when $n_b < 100$; and we found that $n_b < 50$ begins to result in unstable, poor performance. Combined, this suggests a higher $n_b$ is worth it if the total budget is high and desired RMSE is low. Promising directions for future work include deriving an optimal burn-in size per Reviewer MVJi’s suggestion, or developing adaptive burn-in strategies.

---

### Official Review · Reviewer_Pv7c · 2025-11-01

**Soundness:** 3
**Presentation:** 3
**Contribution:** 3
**Rating:** 6
**Confidence:** 2

**Summary:**

This paper develops a theoretical framework for cost-optimal evaluation of generative AI models. It addresses the high cost of using accurate "strong raters" (like humans) by creating policies to actively balance their use with cheap but inaccurate "weak raters" (like model-based autoraters).

Building on prediction-powered inference, the authors derive cost-optimal policies that, given a fixed budget, decide when to pay for the expensive rater to maximize statistical efficiency. The theoretically optimal active policy queries the strong rater most often when the weak rater is most uncertain.

Since the optimal policy's parameters are unknown in practice, the authors test estimation methods like "policy burn-in" (using the first 200 samples) and "policy transfer" (using a related dataset). Experiments on synthetic data and real-world benchmarks (like Chatbot Arena) demonstrate that these methods can achieve the same estimation precision for a fraction of the cost, with the greatest savings seen in tasks with high variability in example difficulty.

**Strengths:**

1. The paper provides a rigorous theoretical framework for active evaluation, extending beyond prior work. Instead of just improving efficiency for a fixed number of expensive annotations, it derives truly cost-optimal policies ($\pi_{random}$ and $\pi_{active}$) that explicitly solve for the best sampling strategy to minimize error given a fixed monetary or computational budget.
2. The work addresses a critical bottleneck in the GenAI lifecycle: the high cost of evaluation. By providing a principled way to combine cheap autoraters with expensive human labels, the framework offers a practical path to achieving high-precision estimates at a much lower total annotation cost.

**Weaknesses:**

1. The theoretically-derived policies, $\pi_{random}$ and $\pi_{active}$, depend on several distributional properties like $Var(H)$, $MSE(H,G)$, and the conditional error $u(x)$. Since these are unknown in a real-world setting, the policies cannot be used out of the box. The paper's practical solutions (burn-in and transfer) are approximations that either require a separate, related dataset or incur an initial "burn-in" cost before any savings can be realized.
2. The benefit of the active policy over the simpler random policy hinges on an accurate estimate of the conditional error, $u(x)$. The paper's own experiments show a significant performance gap between the practical "Active" policy and the "Oracle" policy, which knows the true error. This implies that the current methods for estimating uncertainty are "far from perfect" and are a primary bottleneck limiting the practical gains.

**Questions:**

1. Your practical "burn-in" policy (A2) uses a fixed $n_b=200$ expensive samples to estimate the policy parameters. This initial cost is a critical part of the total evaluation budget. Could you provide a sensitivity analysis showing how the performance of $\pi_{active}$ and $\pi_{random}$ changes for different values of $n_b$? It seems there would be a tradeoff: a small $n_b$ leads to poor parameter estimates, while a large $n_b$ defeats the purpose of saving costs.
2. The main benefit of the active policy over the random one depends on an accurate estimate of the conditional error $u(x)$. You show a significant gap between your "Active" policy and the "Oracle" policy, implying that the $u(x)$ estimates are "far from perfect". For the binary tasks, you used the heuristic $u(x) = G(1-G)$, which assumes the weak rater (G) is a well-calibrated probability. Did you experiment with other methods for estimating $u(x)$ that might be more robust?

---

> ### Author Response · Authors · 2025-11-19
> **Author response**
>
> Thank you for the review! We provide answers to specific questions and remarks (quoted) below.
>
>
> > **Weakness 1:** The theoretically-derived policies... depend on several distributional properties... Since these are unknown in a real-world setting, the policies cannot be used out of the box. The paper's practical solutions (burn-in and transfer) are approximations...
>
>
> **Propositions 1 and 2 are key theoretical results, and we believe they are substantial, new contributions to the theory and understanding of active prediction-powered model evaluation.** That said, yes: they prove that the optimal policies depend on population quantities that are unknown in practice. However, we believe that this theoretical foundation is **valuable in guiding practice,** by telling us what the cost-optimal policies look like and what the important quantities are to estimate. Our experiments in Section 4 indeed provide empirical proof that these oracle policies **can be successfully estimated on data** (e.g., using a small burn-in set or transfer set—and please see our response to subsequent questions about how large the burn-in size $n_b$ needs to be) and that they consistently improve evaluation quality. Not only that, but the theoretical results help us understand when and why our empirical results are (or can be expected to be) much better than classical methods, and, also importantly,  when and why they are not. As discussed in our reply to reviewer MVJi, this is valuable knowledge that is not well explored in any prior literature.
>
> > **Weakness 2:** The benefit of the active policy... hinges on an accurate estimate of the conditional error... The paper's own experiments show a significant performance gap between the practical "Active" policy and the "Oracle" policy...
>
>
> Yes! Our work highlights the importance of good conditional uncertainty estimates for active annotation. More specifically, our paper gives a theoretical foundation for understanding **exactly when, how, and to what extent** having good uncertainty estimates can help (when the variance of the conditional error is high) and when they can’t (when the variance of the conditional error is low). Of course, uncertainty estimation is also an active area of research, but even when using approximations, **our empirical results already show that we can obtain substantial benefits (see Figures 2-3 in the main text, and Figures 6-7 in Appendix D that show that our estimated policies can achieve over 40% reductions in cost on certain tasks).** That said, yes, there is absolutely an empirical gap that remains to the optimal policy in theory—and reducing that gap by improving uncertainty estimation in auto-raters is a profitable, concrete area of future research.
>
>
> > **Question 1:** Could you provide a sensitivity analysis showing how the performance of $\pi_{active}$ and $\pi_{random}$ changes for different values of $n_b$?
>
> We have included an entirely new section in Appendix D.4 that explores the effect of $n_b$ on performance. In general, the policies do not require $n_b$ to be impractically large at all: the parameters quickly converge as $n_b$ grows. For example, taking the Gaussian setting from Section 3, Figure 8 in Appendix D.4.1 explores using a finite sample (i.e., $n_b$ burn-in) to estimate the parameters of both $\pi_{random}$ and $\pi_{active}$ instead of using their oracle settings, for various Var(H) and fixed true MSE(H, G) = Var(U) = 0.5.  **By $n_b = 20$, performance is within 10% of the oracle, by $n_b = 20$  it is within 5%, and by $n_b = 100$ it is well within 2%.** Experiments on the real datasets in Section D.3.2 (where the burn-in set is also used for calibration of G, and warm-starting the estimator) show that even with $n_b$ as low as 50, we can obtain substantial reductions in RMSE at total budgets larger than 50 (which includes the budget for the burn-in set).
>
>
>
> > **Question 2:** Did you experiment with other methods for estimating conditional error $u(x)$ that might be more robust?
>
>
> We chose $u(x) =G(1-G)$ mainly for simplicity. But yes, to work well, this assumes that $G$ is well calibrated. We did use either the transfer data or the burn-in data to calibrate $G$ using Platt scaling, but we agree that exploring more robust estimation methods for $u(x)$ is a great direction for future research. That said, while the $u(x)$ estimate we used isn’t perfect, our **empirical results in Section 4 show substantial gains nonetheless.** Improvements to $u(x)$ are great areas of future work to close the gap between the "Active" and "Oracle" policies.

---

### Official Review · Reviewer_MVJi · 2025-11-01

**Soundness:** 3
**Presentation:** 3
**Contribution:** 3
**Rating:** 6
**Confidence:** 3

**Summary:**

The paper develops a cost-aware framework for hybrid evaluation that mixes a cheap, weak rater with an expensive, strong rater, aiming to estimate the strong rater’s mean judgment under a budget. It derives (i) a closed-form optimal random sampling policy under cost constraints and (ii) an input-adaptive active policy with clipping and a threshold, then studies practical instantiations via policy transfer and burn-in estimation. Experiments show budget savings and reduced MSE compared with always using the strong rater.

**Strengths:**

1. The objective of minimizing estimator error subject to an annotation budget is formalized, yielding a closed-form $ \pi_{\text{random}} $ in terms of costs and weak-rater MSE, and an adaptive $ \pi_{\text{active}} \propto \sqrt{u(x)} $ with principled clipping to respect $ \pi(x)\in(0,1] $ with clear derivation.

2. The transfer and burn-in strategies provide workable recipes, and the paper reports effective budget and cost-savings curves that are easy to interpret.

3. Experiments on real-data seem to match the theory’s qualitative predictions.

**Weaknesses:**

1. The method extends prediction-powered/active inference by optimizing cost-constrained policies and addressing clipping, but much of the estimator form and sequential setup follows prior work.

2. The active policy depends on a non-convex 1-D optimization over $ \tau $, and the paper does not report sensitivity to $ \tau $, mis-estimated $ u(x) $, or misspecified cost ratios, which are likely in practice.

3. The burn-in approach assigns the first $ n_b $ items to the strong rater to estimate parameters, which reduces net gains at small budgets. More discussion on adaptive burn-in size or warm-start reuse across tasks would be useful.

4. In Chatbot Arena experiments, the strong label is also from LLMs, i.e., Gemini 1.5 Flash majority vote. A human-grounded subset would better validate external correctness.

**Questions:**

See Weaknesses.

---

> ### Author Response · Authors · 2025-11-19
> **Author response (1/2)**
>
> Thank you for the review! We provide answers to specific questions and remarks (quoted) below.
>
> > **Weakness 1:** The method extends prediction-powered/active inference by optimizing cost-constrained policies and addressing clipping, but much of the estimator form and sequential setup follows prior work.
>
> **Propositions 1 and 2 are key theoretical results, and we believe they are substantial, new contributions to the theory and understanding of active prediction-powered model evaluation.** Specifically, cost optimality is a significant technical contribution to what has been explored before in active statistical inference, with completely new analysis. In addition, this work carefully lays out a rigorous understanding of when cost-optimal active (i.e., input conditional) annotation is really expected to improve over simple random cost optimal annotation, **and when it is not:** which is something that is not well explored in any prior literature, but is very valuable for the community to know. Our revised draft makes these points clearer.
>
> > **Weakness 2:** The active policy depends on a non-convex 1-D optimization over tau*, and the paper does not report sensitivity to tau*, mis-estimated parameters, or misspecified cost ratios, which are likely in practice.
>
> As noted in Remark 3 (page 4), $\tau$ is a one dimensional scalar, and optimizing it can easily be done with a simple grid search (e.g., over a grid of $k$ different quantiles of $U$ taken from an empirical sample). We have added another experiment in Appendix D.5 that varies the grid size used to search for $\tau^*$ in the same controlled Gaussian setting of Section 3; searching a coarse grid of only ≈ 10 quantiles is sufficient to maximize performance in that setting. With respect to misestimated/specified policy parameters, Appendix B.6 presents bounds on the effects of this on estimator performance. While we do expect approximation error in practice, our empirical experiments in Section 4 show that we can still achieve strong results. We have also added results to Appendix D.4 that show the effects of estimating policy parameters imperfectly with a limited number of empirical samples—see our next response.
>
> > **Weakness 3:** The burn-in approach assigns the first $n_b$ items to the strong rater to estimate parameters, which reduces net gains at small budgets. More discussion on adaptive burn-in size or warm-start reuse across tasks would be useful.
>
> This is a good point to raise! To address it, we have added an entirely new section in Appendix D.4 that explores the effect of $n_b$ on performance. **In general, we find that for smaller $n_b$ we can still achieve substantial improvements with the resulting estimated policies.** For example, taking the Gaussian setting from Section 3, Figure 8 in Appendix D.4.1 explores using a finite sample (i.e., $n_b$ burn-in) to estimate the parameters of both $\pi_{random}$ and $\pi_{active}$ instead of using their oracle settings, for various Var(H) and fixed true MSE(H, G) = Var(U) = 0.5.  By $n_b = 20$, performance is within 10% of the oracle, by $n_b = 20$  it is within 5%, and by $n_b = 100$ it is well within 2%. In terms of net gains, experiments on the real datasets in Section D.4.2 (where the burn-in set is also used for calibration of G, and warm-starting the estimator) show that whenever the total budget is $> n_b$ (where the total budget also accounts for the budget for the burn-in set), we quickly begin to achieve substantial gains over the classical baseline. **This is true even with $n_b$ as low as 50.**
>
> Using an adaptive burn-in size is also a fantastic (but non-trivial) suggestion; we leave it to future work. We have added this suggestion with attribution to anonymous reviewer MVJi to the text in Appendix D.3.2. Warm-start reuse corresponds to our transfer setting for Chatbot Arena (discussed in Section 4.3 and shown in Figure 2): indeed when available, it can be extremely useful.

---

> > ### Author Response · Authors · 2025-11-19
> > **Author response (2/2)**
> >
> > > **Weakness 4:** In Chatbot Arena experiments, the strong label is also from LLMs, i.e., Gemini 1.5 Flash majority vote. A human-grounded subset would better validate external correctness.
> >
> >
> > Agreed; human-grounded scores are a good target estimand. However, we believe that a very valuable perspective our work brings is in expanding the scope of PPI-related methods (which historically have always focused on only human vs. autorater tradeoffs) to any cheap, weak rater with a more expensive, strong rater. LLM autoevaluation has become pretty commonplace in practice (both in academia and industry), so this setup is realistic. Here the Gemini 1.5 Flash majority vote serves as a more accurate and significantly more expensive "strong rater" than the cheap model autorater (Gemma 3B), which is a realistic and relevant cost trade-off in modern LLM benchmarking (Section 1). The Chatbot Arena dataset also is a good example of a setting where Gemma 3B can indeed make some of the same predictions as Gemini 1.5 on some examples, but fail for other, more difficult, examples. Again, the problem we are solving is not necessarily “external correctness”, but rather ensuring strong consistency with any more expensive metric of choice. **We have added additional discussion of this to the draft, please Section 4.2.**
> >
> > That said, we **do also target human-grounded scores** on both the Attributed Question Answering (AQA) and Seahorse tasks, which were in the Appendix D of our submitted draft. With the extra space in this revision, we have moved AQA up to the main paper (Section 4.2-4.3 of the revision).

---

### Author Response · Authors · 2025-11-19
**Main Response Comment**

Thank you to all the reviewers for taking the time to read and review our work. We were pleased to see that reviewers found our work to be “theoretically elegant,” "technically sound,” “clearly articulated,” and addressing a “critical bottleneck in the GenAI lifecycle: the high cost of evaluation.” We also appreciate the comments and questions raised, and have made a significant effort to address them. We have made major changes to the paper, and have uploaded a revised draft with changes in red.

There was one common thread of concern raised by all reviewers, which can be summarized as follows: the theoretical results show that the optimal policies depend on distribution specific quantities which must be estimated in practice---what then is the significance for real world use cases? We have taken this feedback to heart and **have updated the manuscript** to address this in two specific ways:
- **We have revised the Introduction and Contributions sections to more explicitly frame our theoretical results in Propositions 1 and 2 as "oracle" targets that define the necessary ingredients for practical estimation.** We added text clarifying that these theoretical results are essential not just for defining the optimal policy, but for rigorously understanding when cost-optimal active annotation is expected to outperform cost-optimal random annotation (per our analysis in Section 3), which directly informs the design of the practical estimators used in Section 4, and the interpretation of their results.
- **To thoroughly show that these oracle policies can be estimated effectively with limited data, in addition to our existing experiments in Section 4, we have added Appendix D.3-5.** In particular, Appendix D.4 provides analysis on the burn-in size $n_b$, and shows that substantial performance benefits can be obtained over the baseline estimator **with even as few as 50 labeled burn-in examples.** Furthermore, to better highlight the practical utility of this approach, Figures 6 and 7 in Appendix D.3 plot the "Percent Reduction in Cost" relative to the baseline estimator, which show that our **practical approximations can yield over 40% reductions in cost.**

We also made several improvements to the presentation and clarity of the paper. In particular, based on feedback from R-te8E, we have updated the title to be more precise: "Optimal, Active, Prediction-powered AI Model Evaluation on a Budget".

We hope this addresses the main concerns of the reviewers. Additional questions and feedback will be addressed in separate comments.

---

### Meta-Review · Area_Chair_wBN2 · 2025-12-29

**Summary:**

After reading the manuscript, reviewer comments, and authors response, I made my recommendation reject. My major concern lies the clarity. Here are the detailed meta review.

**Research question**

Given an inference target, the authors consider the problem of combining of expensive ground truth labels (e.g., human) and cheaper AI pseudolabels to estimate the target. I agree with one reviewer that the whole pipeline of AI model evaluation is unclear, especially the sequential setting.

**Challenge analysis**

The authors argue that the AI pseudolabels may be imperfect, so they need to use ground truth labels to correct their biases. The core challenge is doing this effectively within a constrained budget. This contrasts with existing solutions (like Zrnic & Candes, and the rest of the prediction-powered inference literature) which do not address budget limitations. Unfortunately, the authors only demonstrate the difference, rather than challenges. It is unclear why Zrnic & Candes' conclusion cannot be generalized to the budget constraint setting, or how difficult to apply Zrnic & Candes' conclusion for the analysis of the budget constraint setting.

**Philosophy**

Since I did not find the target challenge, I did not find the corresponding philosophy, either. In the discussion with authors, the authors provide the description of their solution, rather than solution.

**Solution**

I cited the authors' response to summarize the solution. They use tools from optimization and prediction-powered inference to derive the optimal estimator to achieve this goal. The estimator is unbiased, and strategically leverages the expensive gold labels for the hardest points and the cheap AI pseudolabels for the easiest ones, while staying within cost constraints.

**Theory**

This paper's core contribution lies in the theoretical part. Beyond the unclarity of the setting mentioned above, the theoretical part is also difficult to follow, especially the notation part.

**Experiments**

1. I have to comment the unclarity of the experimental part again.

2. It would like to see in which condition the proposed theory works or does not work. This is lacking in experiments.

3. One reviewer has a concern on the weak experiments. The authors add one more experiment.

**Presentation**

My major concern lies in the presentation. In most places, the authors directly offer the solution without any rationality. The required modification is beyond a minor. I have to give the rejection recommendation.

**Reviewer Concerns:**

Here are outstanding comments.

1. The practicality of the proposed theory. The authors need to further address how to tackle the unknown parameter approximation.

2. Experiments are narrow.

3. Several expressions are vague and fall short of academic standards.

**Reviewer Scores:**

The author response addressed the reviewers' comments to some extend. However, the unclarity of several key points need to be heavily addressed in future versions.

---

### Decision · Program_Chairs · 2026-01-26

Reject